



# Uncertainty of atmospheric microwave absorption model: impact on ground-based radiometer simulations and retrievals

Domenico Cimini[1,2], Philip W. Rosenkranz[3], Mikhail Yu. Tretyakov[4], Maksim A. Koshelev[4], and Filomena Romano[1]

[1]National Research Council of Italy, Institute of Methodologies for Environmental Analysis, Potenza, 85050, Italy
[2]Center of Excellence CETEMPS, University of L'Aquila, L'Aquila, 67100, Italy
[3]Massachusetts Institute of Technology, Cambridge, MA, 02139, USA
[4]Russian Academy of Sciences, Institute of Applied Physics, Nizhny Novgorod, 603950, Russia

*Correspondence to*: D. Cimini (domenico.cimini@imaa.cnr.it)

**Abstract.** This paper presents a general approach to quantify the absorption model uncertainty due to uncertainty in underlying spectroscopic parameters. The approach is applied to radiative transfer calculations in the 20-60 GHz range, which is commonly exploited for atmospheric sounding by microwave radiometer (MWR). The approach however is not limited to any frequency range, observing geometry, or particular instrument. In the considered frequency range, relevant uncertainties come from water vapor and oxygen spectroscopic parameters. The uncertainty of the following parameters is found to dominate: (for water vapor) self and foreign continuum absorption coefficients, line broadening by dry air, line intensity, temperature-dependence exponent for foreign continuum absorption, and line shift-to-broadening ratio; (for oxygen) line intensity, line broadening by dry air, line mixing, temperature-dependence exponent for broadening, zero-frequency line broadening in air, temperature-dependence coefficient for line mixing. The full uncertainty covariance matrix is then computed for the set of spectroscopic parameters with significant impact. The impact of the spectroscopic parameter uncertainty covariance matrix on simulated downwelling microwave brightness temperatures ($T_B$) in the 20-60 GHz range is calculated for six atmospheric climatology conditions. The uncertainty contribution to simulated $T_B$ ranges from 0.30 K (sub-Arctic winter) to 0.92 K (tropical) at 22.2 GHz, and from 2.73 K (tropical) to 3.31 K (sub-Arctic winter) at 52.28 GHz. The uncertainty contribution is nearly zero at 55-60 GHz frequencies. Finally, the impact of spectroscopic parameter uncertainty on ground-based MWR retrievals of temperature and humidity profiles is discussed.

## 1 Introduction

Atmospheric absorption models are used to simulate the absorption/emission of electromagnetic radiation by atmospheric constituents. Atmospheric absorption models are thus crucial to compute the radiative transfer through the Atmosphere (Mätzler 1997; Saunders et al., 1999; Clough et al., 2005; Buehler et al., 2005; Eriksson et al., 2011), which is needed to simulate and validate passive and active remote sensing observations, as from microwave radiometer (MWR) and radar



instruments (Hewison et al., 2006; Maschwitz et al., 2013). Absorption and radiative transfer models, representing the forward operator for atmospheric radiometric applications, are also exploited in physical approaches for the solution of the inverse problem, i.e. the retrieval of atmospheric parameters from remote sensing radiometric observations (Westwater, 1978; Rodgers, 2000; Rosenkranz, 2000; Cimini et al., 2010). Thus, absorption and radiative transfer models, and their
uncertainty, have general implications for atmospheric sciences, including meteorology and climate studies.

Comparisons of different radiative transfer and microwave absorption models have been performed to quantify the difference in calculated brightness temperatures ($T_B$) and the agreement with ground-based, satellite, ship- and air-borne radiometric observations (Westwater et al. 2003; Melsheimer et al., 2005; Hewison, 2006a; Hewison et al., 2006; Brogniez et al., 2016). However, the uncertainty affecting current microwave radiometric observations is often comparable to the
differences in radiative transfer calculations, and thus clear and definite answers were not always obtainable.

Absorption models are based on quantum-mechanics theory and rely on parametrized equations to compute the atmospheric absorption given the thermodynamic conditions and the abundance of constituents (Rosenkranz, 1993). The spectroscopic parameters entering the parametrized equations are determined through theoretical calculations or laboratory and field measurements, and their values are continuously refined (Liebe et al., 1989; Rosenkranz, 1998; Liljegren et al., 2005; Turner
et al., 2009; Mlawer et al., 2012; Gordon et al., 2018; Koshelev et al., 2018). Review papers are published occasionally to summarize the proposed modifications (Rothman et al., 2005; Rothman et al., 2013; Gordon et al., 2017). The absorption models described in Rosenkranz (1998) and Rosenkranz (2017) are cited frequently in this paper and are hereafter called R98 and R17, respectively. The review by Tretyakov (2016) is also cited frequently, meaning Tretyakov (2016) and references therein.

The uncertainty affecting the values of spectroscopic parameters contribute to the uncertainty of the simulated absorption, which in turn affects atmospheric radiative transfer calculations. Thus, the uncertainty affecting spectroscopic parameters contributes to the uncertainty of simulated remote sensing observations, and consequently to the uncertainty of remote sensing retrievals of atmospheric thermodynamic and composition profiles (Boukabara et al., 2005a; Verdes et al., 2005; Long and Hodges, 2012). However, it must be considered that the uncertainty affecting different spectroscopic parameters
may be correlated. Therefore, in addition to the uncertainty affecting the single parameters, the full uncertainty covariance matrix should be estimated for accounting the correlation into radiative transfer calculations and retrievals (Rosenkranz, 2005; Boukabara et al., 2005b).

In the last decade, the Global Climate Observing System (GCOS) Reference Upper-Air Network (GRUAN) has evolved from aspiration to reality (Bodeker et al., 2015). GRUAN is now delivering reference-quality measurement of Essential
Climate Variables (ECV), for which the uncertainty contributions are carefully evaluated. In addition to radiosonde observations (Dirksen et al., 2014), ground-based remote sensing products are planned in GRUAN, including from microwave radiometer (MWR) profilers. Most common MWR profilers operate in the 20-60 GHz range to infer ECV such as tropospheric temperature and water vapor profiles, and vertically-integrated water vapor and liquid water contents. MWR adds value to GRUAN by providing redundant measurements with respect to radiosondes, but covering the complete diurnal




cycle at high (e.g., 1 min) temporal resolution. The various sources of uncertainty for MWR retrievals have been reviewed in the framework of the GRUAN-related GAIA-CLIM project (http://gaia-clim.eu/, Thorne et al., 2017). One such a source is the spectroscopic parameter uncertainty, which appears to be the least investigated among all (Maschwitz et al., 2013; GAIA-CLIM G2.37, 2017). The premises above call for a thorough investigation on the uncertainty affecting spectroscopic parameters entering current microwave absorption models and their impact on MWR simulated observations and retrievals.

Focusing primarily on clear sky retrievals, the main constituents contributing to atmospheric microwave absorption in the 20-60 GHz range are water vapor and oxygen.

Thus, the main purpose of this paper is to introduce a rigorous approach for quantifying the absorption model uncertainty. Although the approach is general and not limited to any particular instrument, observing technique, or frequency range, we demonstrate its use through the application to ground-based microwave radiometer simulations and retrievals. The analysis

thus consists in the following four steps:

(i) review recent work concerning water vapor and oxygen spectroscopic parameters and their associated uncertainties;

(ii) perform a sensitivity study to investigate the dominant uncertainty contribution to radiative transfer calculations;

(iii) estimate the full uncertainty covariance matrix for the dominant parameters;

(iv) propagate the uncertainty covariance matrix to estimate the impact on MWR simulated observations and atmospheric

retrievals.

Thus, the paper is organized as follows: Section 2 summarizes the equations used in the absorption models and defines their parameters; Section 3 presents the results of the uncertainty sensitivity study; Section 4 discusses the approach to estimate the uncertainty covariance matrix; Section 5 presents the impact of spectroscopic uncertainty on simulated downwelling 20-60 GHz $T_B$ and on the associated ground-based atmospheric temperature and humidity profile retrievals; Section 6 presents a

summary, main conclusions, and hints for future work; finally, an appendix reviews recent updates to spectroscopic parameters in the microwave absorption models.

## 2 Review of absorption model equations

Absorption happens when radiation travels through a dissipative medium. The radiation intensity as a function of the path length $l$ through the medium is given by the Beer-Lambert-Bouguer law, $I(l) = I_0 \cdot e^{-\alpha(\nu) \cdot l}$, where $I_0$ is the incident

radiation intensity, $I$ is the transmitted radiation intensity passed through the medium, and $\alpha$ is the absorption coefficient of the medium, which depends on the radiation frequency $\nu$. The absorption coefficient is a macroscopic parameter that represents the interaction of incident electromagnetic energy with the constituent molecules. Here we consider atmospheric absorption, and thus $\alpha(\nu)$ represents the absorption spectrum of the gas mixture forming the atmosphere. The gas absorption spectrum is the sum of two components: the resonant and non-resonant absorption. The resonant absorption is a property of

individual molecules; it occurs at certain frequencies (absorption lines) associated, for example, with the change in the angular momentum of the molecule (rotational transition) or the oscillation frequency (vibrational transition). The non-





resonant absorption arises from the interaction of molecules with each other, i.e. due to the non-ideality of gas. Thus, the gas absorption coefficient can be expressed as the sum of the resonance lines and the non-resonance absorption:

$$\alpha_{total} = \sum \alpha_{line} + \alpha_{non-res} \qquad (1)$$

### 2.1 Resonant absorption

The resonant absorption is modeled computing the contribution of each significant absorption line (line-by-line). Following Rosenkranz (1993), the power absorption coefficient at frequency $\nu$ for a specified molecular species with $n$ molecules per
unit volume is given by:

$$\sum_i \alpha_{line}(\nu, \nu_i) = n \sum_i S_i(T) F(\nu, \nu_i) \qquad (2)$$

where


$$F(\nu, \nu_i) = \frac{1}{\pi}\left(\frac{\nu}{\nu_i}\right)^2 \left[\frac{\Delta\nu_i + Y_i \cdot (\nu-\nu_i)}{\Delta\nu_i^2 + (\nu-\nu_i)^2} + \frac{\Delta\nu_i - Y_i \cdot (\nu+\nu_i)}{\Delta\nu_i^2 + (\nu+\nu_i)^2}\right] \qquad (3)$$

is the line-shape function, while the following line parameters refer to the $i^{th}$ absorption line of the specified molecule: the center frequency ($\nu_i$), the half-width at half amplitude ($\Delta\nu_i$), the integrated intensity at temperature $T$ ($S_i(T)$), and the mixing
parameter ($Y_i$). Note that the summation in Eq. (2) only includes $i>0$, as negative resonances are included in the line-shape function, and the zero-frequency transitions (Debye absorption, which must be taken into account in molecular oxygen), sometimes referred as to $i=0$, is treated below. The line-shape function Eq. (3) does consider that in case two or more lines contribute significantly to the absorption, there may be non-negligible line mixing, in which case the resulting intensity of the band cannot be calculated as a simple sum of isolated line profiles. Instead of that, the line mixing coefficients $Y_i$ account
for the line mixing effect in the first-order (in pressure) approximation suggested by Rosenkranz (1975). A second order expansion was later proposed by Smith (1981), adding coefficients accounting for mixing of line intensities and shifting of line central frequencies.

In the frequency range considered here (20-60 GHz), the line mixing effect is fundamental for understanding the oxygen absorption, while it is negligible for water vapor ($Y_i \cong 0$). Thus, for water vapor, the line-shape function reduces to the van
Vleck–Weisskopf profile, which was demonstrated to fit experimental data best among other analogous shapes (Hill, 1980):

$$F^{VVW}(\nu, \nu_i) = \frac{1}{\pi}\left(\frac{\nu}{\nu_i}\right)^2 \left[\frac{\Delta\nu_i}{\Delta\nu_i^2 + (\nu-\nu_i)^2} + \frac{\Delta\nu_i}{\Delta\nu_i^2 + (\nu+\nu_i)^2}\right] \qquad (4)$$





The van Vleck–Weisskopf profile can also be used for taking into account zero frequency transitions by letting $v_0=0$ (Van

Vleck, 1947). All these transitions are overlapped by each other and can be treated as a single resonance line. This line in $O_2$
may be included in the summation of Eq. (2) as $i=0$, with $v_0=0$, $Y_0=0$. However, a different definition of line intensity must
be used:

$$S'_0(T) = \lim_{v_0 \to 0} \left( \frac{S_0(T)}{v_0^2} \right) \qquad\qquad (5)$$


which has a finite nonzero value as $v_0 \to 0$. Thus, introducing $\gamma_0$ as the $O_2$ zero-line half-width at half amplitude, this
absorption reduces to the following expression, which has the Debye line shape factor (Rosenkranz, 1993):

$$\alpha_0(v, T) = S'_0(T) \frac{n}{\pi} \frac{\gamma_0}{(v^2 + \gamma_0^2)} \cdot v^2 \qquad\qquad (6)$$


Note that the line profiles (3, 4, 6) are valid only when the frequency detuning satisfies $|v - v_c| \ll (2\pi\tau_c)^{-1}$, where $\tau_c$ is the
finite duration of molecular collision. Therefore, a way to model the line absorption is the so-called line wings cut-off, i.e.
assuming zero absorption at detunings larger than a cut-off frequency. The value of the cut-off frequency proposed by
Clough et al. (1989), 750 GHz, is widely accepted and used in some absorption models (R98; Clough et al., 2005). It should

be also mentioned that line profiles (3, 4, 6) take into account only collisional broadening mechanism and ignore additional
line broadening related to thermal molecular movement (Doppler broadening), which has significant effect in the considered
frequency range only at very low gas densities (i.e. altitudes above 60 km). Fine effects of collisional narrowing of resonance
line due to speed dependence of absorbing molecule cross section or velocity - changing collisions are also ignored.

**2.2 Non-resonant absorption**

The non-resonant absorption accounts for the absorption characterized by smooth frequency dependence remaining after
considering the effect of resonant lines. The mechanism for non-resonant absorption arises from the non-ideality of
atmospheric gases and corresponds to the absorption by collisionally interacting molecules. At usual atmospheric conditions
only pair interaction is significant. This interaction during finite time of collision may lead to significant (either positive or

negative) deviation of resonance line far wings from the absorption calculated using profiles (3-6). For each molecule, the
sum of these deviations over all lines gives absorption smoothly varying with frequency. Another component of non-
resonance absorption corresponds to molecular pairs (bimolecular absorption). The latter one can be further subdivided into
three parts, corresponding to free molecular pairs, quasi-bound (metastable) dimers, and true-bound (stable) dimers. All
these absorption contributions are also very smoothly varying with frequency at atmospheric conditions due to either short





life time of bimolecular state (free pairs and quasi-bound dimers) or extremely dense and collisionally broadened spectrum of loosely bound molecular pair (quasi-bound dimers and true-bound dimers).

To model the non-resonance bimolecular absorption in the atmosphere, it should be taken into account that pair interactions occur in any atmospheric gases and their mixtures. For convenience, the treatment of the atmospheric non-resonance absorption is divided in two contributions, one deriving from dry air and the other from water vapor.

The dry contribution is due to the interaction of dry air molecules with each other. Only molecular nitrogen and oxygen are considered, as they account for nearly 100% of the atmospheric mixture and absorption. Because of dominant nitrogen contribution this component can be approximately calculated in the considered frequency range as:

$$\alpha_{dry}(\nu,T) = \alpha_{N_2}(\nu,T)[1 + \varepsilon(\nu,T)] \tag{7}$$


where $\alpha_{N_2}(\nu,T)$ is the absorption due to $N_2$-$N_2$ interactions and $\varepsilon(\nu,T)$ accounts for the absorption due to $O_2$-$O_2$ and $N_2$-$O_2$ interactions, considering $N_2$ and $O_2$ relative abundances and absorption intensities (Boissoles et al., 2003).

Concerning the water vapor contribution to non-resonance absorption, despite the general understanding of the physical nature (e.g. Sine et al., 2012; Tretyakov et al., 2014; Serov, et al., 2017), there are no sufficiently accurate theoretical models

for calculating spectra of all necessary components (especially in gas mixtures) and their temperature dependences. Therefore, for practical purposes parameters of the observed non-resonant absorption are determined using simple empirical models, which have not been supported by accurate theoretical calculations and are based on experimental data only (Tretyakov, 2016). The so-called continuum absorption is thus empirically defined as the difference between the total observed absorption and the calculated contribution of resonance lines:


$$\alpha_{cont} = \alpha_{total} - \sum \alpha_{lines} \tag{8}$$

Note that in such a definition the resulting continuum absorption contains the non-resonant absorption as well as the unknown contribution from resonance line far wings at frequency detunings exceeding the somewhat arbitrary cut-off

frequency introduced above.

### 2.3 Absorption model parameterization

The spectroscopic parameters appearing in the above equations may depend on temperature ($T$) and pressure ($P$). Most experimental data on spectroscopic parameters are obtained near room temperature, and thus tabulated values are available at

reference temperature $T_0$ (usually 296 K or 300 K). Parametric functions are used to express the dependence on $T$ and $P$ in common absorption models.




For the line intensity, the temperature dependence is given by the total number of populated molecular states (the partition sum), which can be calculated numerically (Gamache et al., 2017), and the population of molecular energy levels corresponding to the transition. The latter is calculated from the energy of the lower level and the frequency of the

corresponding transition. Thus, calling $k$ the Boltzmann's constant, $E_{low}$ the energy of the lower level, and $S(T_0)$ the intensity at the reference temperature $T_0$, and introducing the so-called inverse temperature ($\theta = \frac{T_0}{T}$), the intensity is written as (Rosenkranz, 1993):

$$S(T) = S(T_0) \, \theta^{n_S} \exp\left( \frac{E_{low} + h\nu_i/2}{kT_0} (1 - \theta) \right) \tag{9}$$


where the temperature exponent $n_S$ accounts for temperature dependence of the partition sum and differs for asymmetric (e.g. water vapor, $n_S \cong 2.5$) and linear (e.g. oxygen, $n_S \cong 2.0$) molecules.

For pressure-broadened line coefficients, it is convenient to introduce normalized coefficients, relative to the reference temperature $T_0$ and independent of pressure. In general, experimental studies fit them to a function of the form $\gamma =$

$\gamma(T_0) \, \theta^n \, P$, where $\gamma(T_0)$ and $n$ are constant coefficients. The power function is generally suitable for atmospheric applications to account for the temperature dependence of the above parameters as it works well within ±50 K from $T_0$.

For water vapor absorption, the line width and the line center frequency are differently affected in case of broadening induced by water vapor (self broadening, indicated by $s$) or by dry air (foreign broadening, indicated by $a$). Thus, calling $P_w$ and $P_d$ the partial pressures of water vapor and dry air, and $\nu_i^0$ the "zero pressure" transition frequency of the $i^{th}$ absorption

line, the line broadening and shifting are written as, respectively:

$$\Delta\nu_i = \gamma_{i,s}(T_0) \, \theta^{n_{\gamma s}} P_w + \gamma_{i,a}(T_0) \, \theta^{n_{\gamma a}} P_d \tag{10}$$
$$\nu_i - \nu_i^0 = \delta_{i,s}(T_0) \, \theta^{n_{\delta s}} P_w + \delta_{i,a}(T_0) \, \theta^{n_{\delta a}} P_d \tag{11}$$

where $\gamma_{i,s}$, $\gamma_{i,a}$, and $\delta_{i,s}$, $\delta_{i,a}$ are the self and foreign parameters for respectively broadening and shifting at the reference temperature $T_0$, and $n_{\gamma s}, n_{\gamma a}, n_{\delta s}, n_{\delta a}$ are the temperature exponents for line self/foreign broadening and shifting. In R17, the ratio of shift to broadening ($R_i$) is used as a parameter instead of the shifting parameter, e.g. $R_i = \delta_i / \gamma_i$. This implicitly assigns the same temperature dependence to broadening and shifting, which is done because of the absence of relevant measurements for $n_\delta$, although theory suggests that it could differ from $n_\gamma$ (Pickett, 1980).

Similarly, for oxygen it is convenient to introduce normalized broadening ($\gamma_i$) and mixing ($y_i$) coefficients. In addition, the water-to-air broadening ($r_{w2a}$) and mixing ($r'_{w2a}$) ratios are introduced for considering the broadening and mixing of oxygen lines induced by water vapor. Because of the absence of relevant measurements for $r'_{w2a}$, the model assumes $r'_{w2a} = r_{w2a}$. Thus, the width and mixing coefficients are expressed as:





$$\Delta \nu_i = \gamma_i (P_d\, \theta^{n_a} + r_{w2a}\, P_w\, \theta) \tag{12}$$

$$Y_i = (P_d\, \theta^{n_a} + r_{w2a}\, P_w\, \theta)\big(y_i + V_i \cdot (\theta - 1)\big) \tag{13}$$

where $n_a$ is the temperature exponent for oxygen line broadening and $V_i$ are coefficients introduced to account for the $\theta^{n_a+1}$ dependence (Liebe et al., 1992).

Line parameters that most significantly affect the line shape (e.g. $\nu_i$, $S(T_0)$, $E_{low}$, $\gamma(T_0)$, and $\delta(T_0)$) can be found in several spectroscopic databases, e.g. HITRAN (http://hitran.org/, Gordon et al. 2017).

Concerning the water vapor continuum, it has been established (Liebe, et al., 1987; Kuhn et al., 2002; Koshelev et al. 2011; Shine et al., 2012) that the absorption can be represented as two terms corresponding to the interaction of water molecules with each other (self continuum component), and the interaction between water molecules and air molecules (foreign continuum component). In the frequency range considered here, the continuum absorption depends quadratically on frequency (R98) and its temperature dependence is described by a simple exponential function:

$$\alpha_{cont}(\nu, T) = \big(C_s\, \theta^{n_{cs}+3}\, P_w^2 + C_f\, \theta^{n_{cf}+3}\, P_d\, P_w\big) \cdot \nu^2 \tag{14}$$

where we introduced the empirical numerical intensity coefficients for self- ($C_s$) and foreign-induced ($C_f$) water-vapor continuum and their respective temperature-dependence exponents ($n_{cs}$, $n_{cf}$).

For the dry continuum, Rosenkranz et al. (2006) proposed a frequency-dependent factor $f(\nu)$ to fit the data calculated by Borysow and Frommhold (1986), who modelled the bimolecular absorption for $N_2$-$N_2$ pairs. Calling $C_d$ the intensity coefficient of dry-air continuum and $n_d$ the relative temperature-dependence exponent, the dry continuum absorption is modelled as:

$$\alpha_{dry}(\nu, T) = C_d\, f(\nu)\, \theta^{n_d}\, P_d^2\, \nu^2 \tag{15}$$

where the shape of $f(\nu)$ is parametrized in R17 as follows:

$$f(\nu) = 0.5 \cdot \left(1 + \frac{1}{1 + (\nu/450)^2}\right). \tag{16}$$

### 2.4 Atmospheric absorption model in the 20-60 GHz range

In the frequency range considered here (20-60 GHz) and for tropospheric conditions, atmospheric clear air absorption is dominated by oxygen and water vapor. Oxygen produces strong resonant absorption due to transitions in the magnetic dipole





spin-rotation band between 50-70 GHz. Collisional broadening at increased pressures causes the 60-GHz band lines to blend together and at pressures approaching atmospheric and higher the band absorption looks like an unstructured composite feature spreading about ±10 GHz around 60 GHz, and one line at 118.75 GHz. For water vapor, rotational transitions of the electric dipole produce resonant absorption lines extending from the microwave to far infrared range, including lines near

22.235 GHz and 183.31 GHz. Since absorption lines are well separated, the line mixing effect is negligible ($Y_i=0$). In addition to line contributions, water vapor absorption accounts for the continuum component, generally divided into the self and foreign components. More details on the theory of the microwave absorption by atmospheric gases is given by Rosenkranz (1993).

Based on theoretical considerations and laboratory experimental data in the 1960s, the Millimeter-wave Propagation Model

(MPM) was developed for the range from 20 GHz to 1 THz, including the 30 strongest water vapor lines, 44 oxygen lines, and an empirically derived water vapor continuum (Liebe and Layton, 1987). This model has been later revised, modifying the line parameters (Liebe, 1989), the oxygen line coupling (Liebe et al., 1992), the number of water vapor lines and the continuum formulation (Liebe et al., 1993; R98). More details on the differences between these, as well as others absorption models and the comparison with ship-borne, aircraft, and ground-based observations can be found in (Westwater et al., 2003;

Cimini et al., 2004; Hewison, 2006a; Hewison et al., 2006) and references therein. The above models are widely used and have been taken as reference for the last 30 years. For example, the parametrized radiative transfer code RTTOV (Saunders et al., 1999), widely used world-wide to assimilate satellite microwave radiometer observations into weather models, is trained against calculations made with the MPM87 model (Rayer, 2001) and later modifications (Saunders et al., 2017).

Appendix A gives a summary of the modifications to the R98 water vapor and oxygen absorption models proposed in the

open literature in the last 20 years and subsequently imported in the current version of the model (R17). Here, just to show the effects of the adopted modifications, Figure 1 displays the 20-60 GHz downwelling $T_B$ as computed with the R17 model and the difference with respect to the reference R98 model. Six atmospheric climatology conditions have been considered (tropical, midlatitude summer, midlatitude winter, subarctic summer, subarctic winter, U.S. standard).

**3 Sensitivity to uncertainties of spectroscopic parameters**

The atmospheric absorption calculated from a model has in general a nonlinear dependence on some spectroscopic parameters, as reviewed in Section 2. With the assumption of small perturbations, however, one can reasonably linearize that dependence, for a given model:

$$T_B = \mathbf{K}_p \cdot (p - p_0) + T_{B0} \tag{17}$$





where $p$ is a vector whose elements are the parameters in the model, having nominal value $p_0$; $T_B$ is a vector of calculated brightness temperatures at various frequencies using parameter values $p$, while $T_{B0}$ is calculated for parameter values $p_0$, and $\mathbf{K}_p$ represents the model parameter Jacobian, i.e. the matrix of partial derivatives of model output with respect to model

parameters $p$. It follows that the covariance matrix of $T_B$ uncertainties due to absorption model parameter is:

$$\mathbf{Cov}(T_B) = \mathbf{K}_p \mathbf{Cov}(p) \mathbf{K}_p^\top \qquad (18)$$

where the symbol $\top$ indicates transpose matrix. Thus, the full covariance matrix of parameter uncertainties is necessary to

compute the uncertainty of calculated $T_B$, even for just a single frequency. The values of spectroscopic parameters are determined in the spectroscopic literature either theoretically or empirically from field and/or laboratory experimental data, and thus are inherently affected by uncertainty. The uncertainty affecting each spectroscopic parameter contributes to the total uncertainty affecting modelled $T_B$. However, the spectroscopic literature provides at most the uncertainty of individual parameters, not covariance.

Thus, this Section presents a study of the absorption model sensitivity to the uncertainty of spectroscopic parameters, with the purpose of identifying the most significant contributions to the total uncertainty of modelled $T_B$. For the identified relevant parameters, the full covariance matrix is then estimated in Section 4. The approach is as follows. First, the uncertainties affecting spectroscopic parameters are retrieved from published literature or independent analysis. Then, each parameter (or parameter type if known to be highly correlated) has been investigated individually by perturbing its value by

±1-σ uncertainty and computing the impact on the modelled $T_B$. Six different climatologic conditions, as introduced in Figure 1, are considered to account for temperature, pressure, and humidity dependences. Only parameters with 1-σ uncertainty impacting the modelled 20-60 GHz $T_B$ for more than 0.1 K are considered in Section 4 for evaluation of their covariance.

**3.1 Sensitivity to water vapor parameters**

In the 20-60 GHz frequency range under consideration, only two resonant lines (at 22 and 183 GHz) and the continuum contribute non-negligibly to the water vapor absorption. For the model parameters involved with these absorption features, the uncertainties were either retrieved from the spectroscopic literature or, where not available, were estimated from an independent analysis of measurement methods. The resulting uncertainties, as well as nominal values, for the water vapor

parameters considered in this sensitivity analysis are listed in Table 1.

For the resonant absorption, the following parameters are relevant: line frequency ($\nu_i$), intensity ($S_i$) and its temperature coefficient ($n_S$), the lower-state energy ($E_{low}$), air- and water-broadening ($\gamma_a$ and $\gamma_w$) and their temperature-dependence exponents ($n_a$ and $n_w$), and shift-to-broadening ratio ($R_i$). The uncertainty estimates for most of these parameters are given by





Tretyakov (2016) within its review and expert assessment. The only exceptions are the uncertainty estimates for $\gamma_a$, $\gamma_w$, and

$R_i$ at 22 GHz taken from the more recent investigation of Koshelev et al. (2018), and the uncertainty for $n_S$, which has been independently estimated within the 200–400 K temperature range as the maximal difference between numerical calculation of the partition sums at various temperatures published by Gamache et al. (2017) and their power approximation $\theta^{n_S}$.

For the continuum absorption, four parameters are relevant, namely the self- and foreign-induced intensity coefficients and their respective temperature-dependence exponents ($C_s, C_f, n_{cs}, n_{cf}$). Uncertainties for $C_s$ and $C_f$ have been estimated

considering that R17 adopts values adapted from Turner et al. (2009), which also provide the uncertainty estimate for the proposed multiplicative factors (0.79(18) and 1.11(10) respectively for self and foreign coefficients). The uncertainties for $n_{cs}$ and $n_{cf}$ are estimated such to overlap within uncertainty the values given by Koshelev et al. (2011) based on laboratory measurements. The resulting uncertainties (0.6 and 0.8, respectively) are more conservative than those provided originally (Liebe and Layton, 1987; Liebe et al., 1993).

The sensitivity analysis shows that among the nineteen model parameters that were perturbed by the estimated uncertainty (Table 1), only six impact the modelled 20-60 GHz $T_B$ for more than 0.1 K: $C_s, C_f, n_{cf}$, and $S_i$, $\gamma_{i,a}$, $R_i$ at 22 GHz. The sensitivity of 20-60 GHz $T_B$ to perturbations to these six parameters is shown in Figure 2. The impact of both positive and negative perturbations is shown; their symmetry with respect to the zero line suggests that estimated uncertainties represent small perturbations satisfying the linear assumption in Eq. (17). These six parameters are considered in Section 4 for

evaluation of their covariance. Although we note that Tretyakov (2016) indicates larger uncertainty for $n_{cs}$ at temperatures lower than 300 K, it was found that even considering 5-time larger uncertainty (to cover within uncertainty the value given for the range 270-300 K, i.e. 7.6(6)), the impact remains small for the relatively cold climatology. Thus $n_{cs}$ is not considered for the analysis in Section 4.

### 3.2 Sensitivity to oxygen parameters


Oxygen absorption includes zero-frequency band, fine structure spectrum, and pure rotational resonant transitions. The R17 model includes 49 oxygen absorption lines, of which 37 within the 60 GHz band, one at 118 GHz and the remaining 11 in the mm/sub-mm range (200-900 GHz). Uncertainties for the oxygen parameters were either retrieved from the spectroscopic literature or, where not available, estimated from an independent analysis of measurement methods.

For the resonant absorption, the following parameters are relevant: line frequency ($\nu_i$), intensity ($S_i$) and its temperature-dependence exponent ($n_S$), the lower-state energy ($E_{low}$), air-broadening ($\gamma_a$) and its temperature-dependence exponent ($n_a$), normalized mixing coefficient ($y_i$) and its temperature-dependence coefficient ($V_i$), and water-to-air broadening ratio ($r_{w2a}$). The uncertainty estimates for most of these parameters are given by Tretyakov et al. (2005). In particular, Tretyakov et al. (2005) provides frequency uncertainty for 27 lines (N from 1 to 27, where N is the $O_2$ rotational quantum number). For the

other lines, the maximum uncertainty value has been assumed (i.e. 17 kHz), which is conservative with respect to HITRAN.



Resonant line intensities and lower-state energies are taken from HITRAN 2004 database (Rothman et al., 2005). For their uncertainty, we assume 1% and 0.25%, respectively. The latter is a rather conservative estimate, though its contribution turned out to be irrelevant. Note that the 1% uncertainty in $O_2$ line intensities is considered to originate mainly from the uncertainty of experimental measurements of electronic transitions band integrated intensity, which were used for intensity

calculations of microwave lines. This uncertainty should be correlated for all lines by principle of determination and thus we assume a single variable affecting all the lines. Uncertainty of $n_S$ value for the 200–350 K temperature range was evaluated the same way as for water vapor lines, i.e. comparing partition sum calculations by Gamache et al. (2017) with their power-law approximation.

Values for oxygen line air-broadening and mixing parameters are taken from Tretyakov et al. (2005). Line broadening

parameters are measured through low-pressure laboratory experiments. Since individual lines are isolated at low pressures, no correlation is considered between parameters of different lines. Mixing parameters are determined at higher pressures, and their values are correlated with the previously determined low-pressure parameters. So, the line-mixing parameters are correlated with both themselves and the line air-broadening parameters. Because of this relationship, consistency requires that the number of considered line-widths and the number of considered mixing coefficients should be the same. Tretyakov

et al. (2005) derived mixing coefficients for line with N from 1- to 33+ (34 in total), then extrapolated to lines with N>33 (i.e. four weak lines of the 60-GHz complex). Thus, we firstly investigated the impact of these remaining four and the eleven rotational higher frequency lines on to 20-60 GHz $T_B$ by considering conservative and completely correlated uncertainty estimates (10% for line broadening and 20% for line mixing parameters). The impact was found to be negligible (<0.1 K) and thus these fifteen lines are not further considered in the following analysis. For the remaining thirty-four lines (N from 1-

to 33+), the uncertainty for line air-broadening, mixing, and mixing temperature-dependence coefficients is evaluated through the full covariance matrices, so their treatment is postponed to Section 4.

For the air-broadening temperature dependence coefficient, R17 retains a uniform value (0.8) for all lines (Liebe, 1989). We assume 0.05 uncertainty, which covers more recent measurements from Makarov et al. (2008) and Koshelev et al. (2016). Since R17 adopts the water-to-air broadening ratio $r_{w2a}$, its value and uncertainty are estimated respectively as the mean and

standard deviation calculated by Koshelev et al. (2015) from a set of 19 measurements (N from 1 to 19).

For the zero-frequency absorption, two parameters are relevant, the intensity ($S'_0$) and broadening ($\gamma_0$) of the pseudo-line. The intensity of the zero-frequency absorption is from the JPL catalogue. For the zero-frequency line broadening, consideration of the measurements cited in Danese and Partridge (1989), as well as those of Ho *et al.* (1972) and Kaufman (1967), lead us to assign an uncertainty of 50 MHz/bar to the absorption model's value of $\gamma_0$ = 560 MHz/bar at 300 K. Note

that uncertainties in the intensity and broadening coefficients of the zero-frequency component are negatively correlated, because it is very difficult to measure the broadening independently of the intensity for this pseudo-line. This estimate based on the spread of published measurements accounts for the combination of intensity and broadening uncertainties.

The sensitivity analysis shows that among the model parameters in Table 2, which were perturbed by the estimated uncertainty, only the following impact the modelled 20-60 GHz $T_B$ for more than 0.1 K: $S_i$, $\gamma_a$, $n_a$, $y_i$, $V_i$, $\gamma_0$. The sensitivity




of 20-60 GHz $T_B$ to perturbations to these parameters is shown in Figure 3. As for water vapor, the impact of positive and negative perturbations is symmetric with respect to the zero line, suggesting that the linear assumption is valid for the estimated uncertainties. Note that the perturbation to $S_i$ and $n_a$ affect all lines simultaneously, while the other resonant line parameters have been perturbed line by line. Although for the present ground-based application the uncertainty of only a few lines is relevant, we prefer to keep all the 34 to make the calculation of the parameter uncertainties more generally useful

(e.g. for satellite observations). Thus, the above six parameters ($S_i$, $\gamma_a$, $n_a$, $y_i$, $V_i$, $\gamma_0$) are considered in Section 4 for evaluation of their covariance. While for $S_i$, $n_a$, and $\gamma_0$ we consider three scalar parameters, for $\gamma_a$, $y_i$, and $V_i$ we consider 34 lines (N from 1- to 33+), leading to 34 coefficients for each parameter type.

## 4 Estimation of uncertainty covariance matrix

The sensitivity analysis of Section 3 shows that the absorption model uncertainty on downwelling 20-60 GHz $T_B$ is dominated by the uncertainty on 6 spectroscopic parameters for water vapor and up to 105 parameters for oxygen. For these parameters, we require the full covariance matrix of parameter uncertainties to compute the uncertainty of calculated $T_B$ at any given frequency. This section summarizes the methods used to estimate the uncertainty covariance matrix, including the off-diagonal terms giving the covariance of each parameter with the others. Additional details can be found in Rosenkranz *et*

*al.* (2018) (abbreviated as R18 below). However, the analysis here differs in three respects from the preliminary version in R18: in the method of estimating $\mathrm{Cov}(C_f, C_s)$, in use of a smaller uncertainty for $\gamma_0$, and inclusion of $\mathrm{Cov}(\gamma_0, n_a)$ which was neglected in R18.

Although we use different methods to estimate covariances depending on how the parameter values were measured, some general principles apply. If a set of variables $a_i$ have a causal dependence on another set of variables $b_k$,


$$\Delta a_i = \sum_k (\partial a_i / \partial b_k)\, \Delta b_k, \tag{19}$$

and the *b*'s have an uncertainty covariance matrix **Cov**(*b*), then

$$\mathrm{Cov}(a_i, b_m) = \langle \Delta a_i\, \Delta b_m \rangle = \sum_k (\partial a_i / \partial b_k)\, \mathrm{Cov}(b_k, b_m), \tag{20}$$

where the angle brackets denote expectation value, and the *b*'s contribute an amount





$$\Delta \text{Cov}(a_i, a_j) = \sum_m \text{Cov}(a_i, b_m)\,(\partial a_j / \partial b_m) \tag{21}$$


to the uncertainty covariance of the $a$'s. There may also be other contributions to **Cov**($a$).

A probability distribution can be conditional, and the uncertainty of one parameter may be conditioned on an assumed value for a different parameter. Sometimes reported values of a parameter or set of parameters have been adjusted to fit measurements, while the experimenters considered other relevant spectroscopic parameters fixed. Now if we wish to include

in our analysis the uncertainty of one of the latter parameters ($b$), and it has a covariance with a fitted parameter $a$, the influence of $b$ on $a$ will increase the uncertainty of $a$ above that which was found in the original experiment. That increment of variance is also given by Eq. (21), which in the scalar case is equivalent to

$$\Delta(\sigma_a^2) = [\text{Cov}(a,b)/\sigma_b]^2. \tag{22}$$


### 4.1 Uncertainty covariance matrix for water vapor parameters

Section 3.1 shows that for water vapor absorption six spectroscopic parameters dominate the uncertainty of modelled 20-60 GHz $T_B$: three related with the continuum ($C_s$, $C_f$, $n_{cf}$) and three with the 22 GHz resonant line ($S_i$, $\gamma_{i,a}$, $R_i$). Sections 4.1.1-4.1.3 describe the methods used to estimate the covariances of these six water vapor spectroscopic parameters. Although the

covariance matrix is the basic object needed for calculation, Table 3 lists both the estimated covariances of water-vapor parameter uncertainties and the corresponding correlation coefficients, because the latter are more easily comprehended, being pure numbers and normalized to the interval (-1,1). The numerical values of the full covariance matrix are also provided as a supplementary data file (in ASCII and netCDF formats).

### 4.1.1 Covariance between water-vapor line parameters


Intensity, width and shift affect a line profile in different ways. But even if the original spectroscopic measurements covered the line profile adequately, a noticeable negative correlation between width and intensity arises if both are simultaneously estimated from measured absorption. In the present case, the only water line that survived the sensitivity screening for the 20-60 GHz band is the one at 22.2 GHz; the intensity used here was calculated independently from the width, and the width was

measured without using that intensity. Therefore, we consider errors in those two parameters to be uncorrelated. However, the absorption model code under investigation here (R17) uses the aforementioned ratio of shift to width ($R = \delta_a/\gamma_a$, where $\delta_a$ and $\gamma_a$ are respectively the shift and width coefficients). As shown in R18, that introduces a covariance between $R$ and $\gamma_a$ of



$$\text{Cov}(R, \gamma_a) = -\sigma_{\gamma a}^2 R/\gamma_a \tag{23}$$


where $\sigma_{\gamma a}^2$ is the uncertainty variance of $\gamma_a$, and it corresponds to the small correlation of +1% shown in Table 3 (positive because the nominal value of $R$ is negative for this line).

### 4.1.2 Covariance between $C_f$, $C_s$ and other water-vapor parameters

By definition, the water-vapor continuum is the remainder after the contribution of local resonant lines has been subtracted. Thus, if a line width is revised, the continuum should also be revised to compensate and reproduce as well as possible the original brightness-temperature measurements of Turner *et al.* (2009) from which the continuum was derived. That was done in adjusting the continuum coefficients $C_f$ and $C_s$ for use with updated line parameters in R17. It should be the case no matter which line is revised. If we separate the model parameters into continuum (*con*) and line types, then as discussed in R18, the

above statements are equivalent to requiring that for each line separately, the covariance between the continuum and line parameters and the line-parameter covariance matrix satisfy:

$$\mathbf{K}_{p_{con}}\text{Cov}(p_{con}, p_{line}) + \mathbf{K}_{p_{line}}\text{Cov}(p_{line}) = 0. \tag{24}$$

In order for the above equation to hold over a range of humidity, it should apply to self- and foreign-gas effects separately. Both $R$ and $\gamma_a$ apply to dry air, so we set $\text{Cov}(C_s, R)=0$ and $\text{Cov}(C_s, \gamma_{i,a})=0$. On the other hand, line intensity S affects both components of the continuum, with resulting covariances; then Eq. (24) can be solved for $\mathbf{Cov}(p_{con},S)$ by making $\mathbf{K}_{p_{con}}$ 2x2 (see R18). As shown in Table 3B, the correlations of the continuum parameters with the 22-GHz line parameters are very small, because this is one of the weaker water lines. If our matrix had included parameters for the 183-GHz water line, their

covariances with the continuum might well be significant.

Although n<sub>cf</sub>, the continuum foreign broadening temperature exponent, is not a line parameter, it was held fixed by Turner *et al.* (2009) in fitting $C_f$ and $C_s$ to the measured $T_B$. Therefore, any subsequent change in n<sub>cf</sub> should require a compensating change in $C_f$; hence, from Eq. (24)

$$\text{Cov}(C_f, n_{cf}) = -\text{K}_{n_{cf}}\sigma_{n_{cf}}^2/\text{K}_{C_f}, \tag{25}$$

which turns out to produce a significant covariance (Table 3A). If $C_f$ is thus compensated, $C_s$ should not change, so $\text{Cov}(C_s, n_{cf}) = 0$.



### 4.1.3 Covariance between $C_f$ and $C_s$

For the water vapor continuum, R17 adopts the multipliers proposed by Turner *et al.* (2009) to the R98 parameter values of $C_f$ and $C_s$, with small re-adjustments to accommodate the updated line widths in R17. Turner *et al.* (2009) derived the multipliers by adjusting them to fit ground-based radiometer measurements at 150 GHz. The simultaneous fitting of two coefficients results in a correlation between them.

When brightness-temperature measurement errors are uncorrelated, with variance $\sigma_n^2$, a least-squares fit (see, e.g., van der Waerden, 1969; Stuart and Ord, 1991) results in the parameter-error covariance matrix

$$\mathbf{Cov}(C) = <\Delta C \Delta C^{\mathsf{T}}> = \sigma_n^2 \acute{K}^{-1} = \sigma_n^2\, adj(\acute{K})/det(\acute{K}) \qquad (26)$$

in which $C$ is a vector containing the elements $C_f$ and $C_s$, and $\acute{\mathbf{K}}$ is a matrix with elements

$$\acute{K}_{ij} = \sum_m (\partial T_{Bm}/\partial C_i)\,(\partial T_{Bm}/\partial C_j) \qquad (27)$$

where $m$ is the index for the measurements of $T_B$ and indexes $i$ and $j$ equal 1 for $C_f$ or 2 for $C_s$; the derivatives are to be evaluated for each atmospheric profile corresponding to $T_{Bm}$, at the fitted values of $C_f$ and $C_s$. When the correlation coefficient $\rho_{fs}$ between $C_f$ and $C_s$ uncertainties is evaluated from Eq. (26), $\sigma_n^2$ cancels, as does the determinant except for its sign, which in this case is positive. Thus, for the simple case of the 2x2 matrix,

$$\rho_{fs} = -\acute{K}_{12}\left(\acute{K}_{22}\acute{K}_{11}\right)^{-\frac{1}{2}}. \qquad (28)$$

Although Turner *et al.* (2009) do not give the correlation coefficient, it can be estimated from a simulation covering the same range of integrated water-vapor content, 0.37 to 2.76 cm. We used twelve values of humidity distributed over this range, in a subarctic-summer model atmosphere, yielding $\rho_{fs}$= -0.87, which is (presumably) approximately what Turner *et al.* would have calculated. Then using the experimentally-determined uncertainties from Table 1, we have $Cov(C_f, C_s)$=-1.57·10$^{-19}$ (which is ~11% larger than previously estimated using a more indirect method in R18).




Turner *et al*. (2009) held other parameters constant while adjusting the continuum coefficients $C_f$ and $C_s$. When we introduce variance of $n_{cf}$ and its covariance with $C_f$ (see section 4.1.2), then as discussed in reference to Eq. (22), a corresponding increase by $[Cov(C_f,n_{cf})/\sigma_{ncf}]^2$ to the experimentally-determined variance of $C_f$ is required. That increases $\sigma_{Cf}^2$ from $3.09\cdot10^{-21}$ to $4.58\cdot10^{-21}$, which is the value in Table 3A. However, when **Cov**(T$_B$) is computed this increased variance will be offset by

the negative contribution of $Cov(C_f,n_{cf})$. (Had $n_{cf}$ been included in the least-squares fit, then $\acute{K}$ would have been a 3x3 matrix, which would have produced a different result originally.) Variance contributions from the 22-GHz line parameters are negligible. The correlation coefficients in Table 3B were then computed using the modified value of $\sigma_{Cf}$.

**4.2 Uncertainty covariance matrix for oxygen parameters**

The sensitivity analysis in Section 3.2 shows that for oxygen absorption six spectroscopic parameter types dominate the uncertainty of modelled 20-60 GHz T$_B$: line intensity ($S_i$), air-broadening ($\gamma_a$) and its temperature-dependence exponent ($n_a$), normalized mixing coefficient ($y_i$) and its temperature coefficient ($V_i$), and zero-frequency broadening ($\gamma_0$). Parameters $n_a$ and $\gamma_0$ are scalar, while $\gamma_a$, $y_i$, and $V_i$ are vectors of 34 components (for lines with N from 1- to 33+); albeit $S_i$ is also a vector, its percent-uncertainty is a scalar, thus leading to a 105x105 uncertainty covariance matrix. Sections 4.2.1-4.2.4 describe the

method used to estimate the uncertainty covariance of these 105 oxygen spectroscopic parameters with respect to each other. The numerical values of the full covariance matrix are provided as a supplementary data file (both in ASCII and netCDF formats). Figure 4 depicts the resulting matrix as a color-scale image of sign-adjusted correlation coefficients. For any two parameters $p_1$ and $p_2$, with nominal values $\acute{p}_1$ and $\acute{p}_2$ and correlation coefficient $\rho(p_1,p_2)$, the sign-adjusted correlation is defined as:


$$\rho_{SA}(p_1,p_2) = sign(\acute{p}_1)sign(\acute{p}_2)\rho(p_1,p_2) \tag{29}$$

If $\acute{p}_1$ and $\acute{p}_2$ have the same sign, $\rho_{SA}(p_1,p_2)$ reduces to $\rho(p_1,p_2)$. If the signs differ, then $\rho_{SA}(p_1,p_2)$ has sign opposite to $\rho(p_1,p_2)$. If the standard deviations are small compared to the nominal values, as generally is the case here, $\rho_{SA}(p_1,p_2)$

gives the correlation between the absolute values of the parameters. $\rho_{SA}(p_1,p_2)$ can be negative, as is the case for the relation between line intensities and the mixing coefficients, which indicates that a positive error in intensities results in underestimation of line mixing.



### 4.2.1 Covariance between oxygen-line broadening coefficients

Values for oxygen line air-broadening are taken from Tretyakov *et al.* (2005). They measured $N_2$-broadening of $O_2$ lines with rotational quantum numbers N from 1 to 19 and self-broadening for N from 1 to 27 (the 1- line had previously been measured in Tretyakov *et al.* (2004)). Uncertainties of the measured line widths were estimated here by considering the results of Tretyakov *et al.* (2005) and Koshelev *et al.* (2016) together. Three sources were assumed to contribute to the error budget: (*i*) the statistical uncertainty was determined from a Pade-approximation (Koshelev *et al.*, 2016) of the N-

dependence of $\gamma_a$, weighting each of the data by its respective $1/\sigma$; (*ii*) pressure gauge uncertainty of 0.25%; and (*iii*) uncertainty 0.5 C of the temperature sensors. The total uncertainty for each line's air-broadening was determined as root-sum-of-squares. Uncertainties calculated for all lines with $N \leq 19$ are close to each other at ~0.014 GHz/bar, so we use this value for all lines with $N \leq 19$. Even though the lines were measured separately by Tretyakov *et al.* (2005), the pressure-sensor and temperature-sensor uncertainties contain systematic components which (due to the same experimental setup) may

have introduced minor correlations between line widths. However, the broadening parameter uncertainty originates mainly from the unknown baseline of the apparatus. The work by Koshelev *et al.* (2016), in which different sensors were used, confirmed that there was no noticeable bias in the earlier measurements. This reasoning allows us to neglect potential correlations of the measured line widths.

For the remaining lines, Tretyakov *et al.* (2005) extrapolated the broadening coefficients by a straight-line graphical method,

assuming a pivot value (hereafter indicated with subscript *) such that

$$\gamma_N = \gamma_* + (N - N_*)\mu \qquad (30)$$

where $N_*$=11 for $N_2$-broadening and 17 for pure $O_2$; $\mu$ is the slope of the straight line and $\gamma_*$ averages the N- and N+ lines for

$N_*$. The extrapolation introduces correlations among those coefficients and between them and the measurements with $N>N_*$ which were used to determine the straight line, as discussed in detail in R18. Also, the uncertainties of the extrapolated broadening coefficients increase with N, up to a maximum of 0.032 GHz/bar at N=33. For the purpose of estimating covariances, the extrapolation was modeled as though it was a formal linear regression. This assumes that a straight line is the right extrapolation method, which seems reasonable, although it cannot be tested because the very weak lines have not

been measured.

Fig. 4 represents the sign-adjusted correlation coefficients as a color image. The extrapolated coefficients (#24-37 in Fig. 4) are strongly correlated among themselves, although not perfectly. On the other hand, the uncertainty of the zero-frequency broadening coefficient (#3) is assumed to be uncorrelated with the line air-broadening uncertainties. Fig. 5 shows the $\gamma_a$ values given by Tretyakov *et al.* (2005) and the associated uncertainties as estimated above, together with the values and the

uncertainties of y and V, which are treated in the next two sections.




### 4.2.2 Covariance between oxygen line mixing coefficients

Values for oxygen line mixing coefficients are taken from Tretyakov et al. (2005), where mixing coefficients were determined from measurements made near one-atmosphere pressure and temperatures near 22-24 C, by an algorithm which makes them

dependent on the other parameters. Hence, uncertainties in those other parameters contribute uncertainties to the mixing coefficients as well as correlations with them. R18 shows that the estimation algorithm can be represented in the form of a vector equation:

$$y = \mathbf{A}(\alpha - \alpha^b) + b \tag{31}$$


where $y$ is the vector of normalized mixing coefficients defined by Eq. (13), $\mathbf{A}$ is the matrix representing the linear estimation operation, $\alpha$ is the vector of absorption measurements, and $\alpha^b$ is a vector of absorption calculated from a baseline mixing-coefficient set $b$. Hence, applying Eqs. (20-21),

$$\mathbf{Cov}(y) = \sigma_{noise}^2 \mathbf{A}\mathbf{A}^\top + (\mathbf{I} - \mathbf{A}\mathbf{K}_y)\mathbf{Cov}(b)(\mathbf{I} - \mathbf{A}\mathbf{K}_y)^\top + \mathbf{A}\mathbf{K}_\gamma \mathbf{Cov}(\gamma)(\mathbf{A}\mathbf{K}_\gamma)^\top + \sigma_S^2 \mathbf{A}\,\alpha^b(\mathbf{A}\,\alpha^b)^\top, \tag{32}$$

where $\mathbf{I}$ is the identity matrix, $\mathbf{K}_y$ and $\mathbf{K}_\gamma$ are matrices of partial derivatives of baseline absorption with respect to y and γ, respectively, and $\sigma_S$ is the fractional uncertainty in line intensities. The first term above is the contribution of measurement noise with variance $\sigma_{noise}^2$ and the third and fourth terms represent the uncertainty contributed by line widths and intensities in

the derivation of the y's. In Tretyakov *et al*. (2005), the baseline mixing coefficients were taken from Liebe *et al*. (1992), which derived them by essentially the same algorithm, with very similar smoothing characteristics. Therefore, in the second term of Eq. (32), the projection operator $(\mathbf{I} - \mathbf{A}\,\mathbf{K}_y)$ should remove the variation of the mixing coefficients obtained in Liebe *et al*. (1992), and the only part that will survive is the original baseline, which is attributable to the coupling between the positive-frequency resonances and the negative-frequency and zero-frequency bands. In R18, the contribution of the second

term in Eq. (32) is estimated as $(\sigma_{\gamma_0}/\nu_b)^2$ to each element of $\mathbf{Cov}$(y), with $\nu_b$= 40 GHz.

The mixing coefficient of the 1- line was measured separately in Tretyakov *et al*. (2004), so it is not correlated with the others. Their estimated uncertainty for its value is $\sigma_y(1-) = 0.01$ bar$^{-1}$. The y-values measured at 295K in Tretyakov et al. (2005) were adjusted to 300K using the temperature coefficients given by Liebe et al. (1992). However, for the sake of simplicity that small correction was ignored here, and the uncertainties of mixing coefficients at $T_0$=300 K are considered to be the same as

the measured coefficients. Hence, we assume no correlation between the line mixing coefficients at 300K and the line-mixing temperature coefficients, since they originate from different laboratories.



### 4.2.3 Covariance between oxygen line-mixing temperature coefficients

First-order line mixing parameterization in R17 is given by Eq. (13). Table 5 of Tretyakov *et al.* (2005) lists coefficients $a_5$ and
$a_6$ for each line, a notation retained from Liebe *et al.* (1992). These are related to the line mixing coefficients as $y_i=a_5+a_6$, and
temperature coefficients as $V_i=a_6$. Liebe *et al.* (1992) measured line mixing at three temperatures and determined $a_6$ by a linear
regression versus $\theta$.  We calculate the covariance matrix for the V's as

$$\mathbf{Cov}(V) = \sum_k x_k^2 \sigma_{noise}^2(T_k)\, \mathbf{A}(T_k)\mathbf{A}^\mathsf{T}(T_k) + \sigma_{na}^2 [\partial V/\partial n_a][\partial V/\partial n_a]^\mathsf{T} + \varepsilon_{V\,sys}\varepsilon_{V\,sys}{}^\mathsf{T} \qquad (33)$$


where $x_k$ is the influence given by the regression to the mixing coefficients at $T_k$ in determining the V's (see R18). The
baseline $b$ doesn't contribute to $V$, because the three values of $x_k$ sum to zero. The first term in Eq.(33) is the measurement-
noise contribution. Unlike the model parameters that are defined at 300K, the V-coefficients depend on the value of $n_a$, and its
uncertainty $\sigma_{na}$ contributes the second term in Eq.(33); the derivatives $\partial V_i/\partial n_a$ were evaluated by finite differences. The third
term in Eq.(33) results from comparison of Liebe *et al.* (1992) to later work, which indicates that it contained some systematic
errors in intensities (generally ~1% or less) and in line widths (typically ~3.3% smaller than those measured in Tretyakov *et
al.*, 2005). The effect on $V$ of those systematic errors, $\varepsilon_{Vsys}$, was also evaluated numerically, as described in R18. We combine
systematic and random errors in Eq.(33), following the practice recommended by JCGM (2008).

### 4.2.4 Covariance between different oxygen parameter types

The discussion in connection with Eqs. (20) and (21) indicates that corresponding to the second, third, and fourth terms in Eq.
(32) for $\mathbf{Cov}(y)$, there must be uncertainty covariances between the line mixing coefficients of the 60-GHz band and the line-
width and intensity parameters:

$\mathrm{Cov}(y, \gamma_0) = -\sigma_{\gamma_0}^2 [\nu_b{}^{-1} + \mathbf{A}\, K_{\gamma_0}]$       (34)

$\mathbf{Cov}(y, \gamma_a) = -\mathbf{A}\, K_\gamma\, \mathbf{Cov}(\gamma_a)$       (35)

$\mathrm{Cov}(y, S) = -\sigma_S^2\, \mathbf{A}\, \alpha^b.$       (36)

The negative signs in these equations originate because the computed baseline absorption occurs with a minus sign in the
determination of the y-coefficients. Likewise, corresponding to the second term of Eq. (33) for $\mathbf{Cov}(V)$, there is an uncertainty
covariance between the V-coefficients and $n_a$:





$$\mathrm{Cov}(V, n_a) = \sigma_{na}^2 \, \partial V / \partial n_a. \tag{37}$$

The value of $\gamma_0$ was determined by Danese and Partridge (1989) from radiometer measurements of the sky at a mountain site. Because the atmospheric emission depends on the temperature profile, a covariance with $n_a$ results. We calculate a typical value for that site (White Mountain) of $K_{na}/K_{\gamma0} = 0.10$ GHz/bar; thus, in analogy with Eq. (25),

$$\mathrm{Cov}(\gamma_0, n_a) = -(K_{na}/K_{\gamma0}) \, \sigma_{na}^2 = -2.5 \cdot 10^{-4} \text{ GHz/bar}, \tag{38}$$


corresponding to $\rho(\gamma_0, n_a) = -0.10$. The increment of uncertainty-variance for $\gamma_0$ due to Eq. (38) is two orders of magnitude smaller than the value assigned to $\sigma_{\gamma0}^2$, and therefore negligible.

## 5 Uncertainties propagation to ground-based brightness temperature and retrievals

The uncertainty covariance matrices estimated in Section 4 for water vapor and oxygen spectroscopic parameters are combined together to form $\mathbf{Cov}(p)$, a 111 x 111 matrix. The two matrices are combined block-diagonally, i.e. assuming no cross-covariances between $H_2O$ and $O_2$ absorption model parameter uncertainties. Thus, $\mathbf{Cov}(p)$ represents the uncertainty covariance matrix of $H_2O$ and $O_2$ absorption model parameters that were judged relevant for downwelling $T_B$ in the 20-60 GHz range. In this Section, $\mathbf{Cov}(p)$ is propagated to estimate its impact on simulated downwelling $T_B$ and ground-based
temperature and humidity retrievals.

### 5.1 Uncertainty on simulated brightness temperatures

The propagation of the absorption model parameter uncertainty to calculated $T_B$ is given by Eq. (18), which requires the knowledge of $\mathbf{K}_p$, i.e. the Jacobian of calculated $T_B$ with respect to model parameters. The Jacobian $\mathbf{K}_p$ is a $n_{freq}$ x $n_{par}$ matrix,
where $n_{freq}$ is the number of frequency for which the $T_B$ uncertainty should be calculated and $n_{par}$ is the number of considered parameters, 111 in our case. Here we set $n_{freq}$=437, which includes 401 equally-spaced frequencies from 20 to 60 GHz (by 0.1 GHz increment), plus 36 corresponding to the central frequencies of two widely-deployed commercial MWR, i.e. the HATPRO (Rose et al., 2005) and MP-3000A (Ware et al., 2003). The Jacobian $\mathbf{K}_p$ has been estimated numerically by perturbing each parameter individually by a small amount (corresponding to the parameter 1-$\sigma$ uncertainty). To represent
different climatology conditions, six realizations of $\mathbf{K}_p$ have been computed using the six atmospheric climatology





conditions introduced in Figure 1. Thus, $\mathbf{Cov}(T_B)$ is computed from Eq. (18) using $\mathbf{Cov}(p)$ and $\mathbf{K}_p$ estimated as above. Figure 6 reports $\sigma(T_B)$, that is the square root of the diagonal terms of $\mathbf{Cov}(T_B)$, for the whole 20-60 GHz range and for the six atmospheric climatology conditions. Similarly, $\sigma(T_B)$ values at the central frequencies of the two commercial MWR are reported in Table 4 (HATPRO, 14 channels) and Table 5 (MP-3000A, 22 channels).

To appreciate the dominant contributions within the frequency range, the different parameters have been grouped in seven types: intensity S (for both $O_2$ and $H_2O$), $O_2$ line width $\gamma_a$, $O_2$ zero-frequency line width $\gamma_0$, $O_2$ line mixing ($y$), $O_2$ line mixing temperature dependence ($V$), $H_2O$ continuum, $H_2O$ line width $\gamma_a$ and shift-to-width ratio $R$. The contribution of each type to $T_B$ uncertainty was estimated by propagating the uncertainty covariance matrix reduced to the size of parameters belonging to that type only. Figure 7 shows the resulting contributions computed for the tropical climatology conditions. We choose

tropical conditions so that features at 22.2 GHz are evident above the continuum absorption.

Thus, looking at Figures 6-7 and Tables 4-5, it seems convenient discussing the 20-60 GHz range in four parts: the proximity of 22.2 GHz water vapor line (20-26 GHz), the atmospheric window (26-45 GHz), the low-frequency oxygen wing (45-54 GHz), and the opaque oxygen band (54-60 GHz). In the following, the contribution dominance is inferred from Figure 7, while the typical values from Figure 6 and Tables 4-5:

- 20-26 GHz: $T_B$ uncertainty is dominated by uncertainty in water vapor line width and shift coefficients, going from ~0.3 K (sub-Arctic winter) to nearly 1.0 K (tropical).

- 26-45 GHz: $T_B$ uncertainty is dominated by uncertainty in water vapor continuum parameters, increasing with frequency from ~0.4 K to 1.2 K, with ~0.2 K larger uncertainty in tropical with respect to other climatology conditions.

- 45-54 GHz: $T_B$ uncertainty is dominated by uncertainty in oxygen line-mixing parameters (up to 2 K). Water vapor continuum, line mixing temperature dependence, and line intensity parameters also contribute to a lesser extent (up to 1.0-1.2 K) at respectively increasing frequency. The total $T_B$ uncertainty decreases with increasing temperature, that is lower for tropical (up to 2.7 K) then for sub-Arctic winter (up to 3.4 K) conditions.

- 54-60 GHz: $T_B$ uncertainty is below 0.5 K at 54-55 GHz and rapidly approaches zero for frequencies above 55

GHz. In this very opaque region, the contribution of absorption model parameters to simulated ground-based $T_B$ is negligible.

The qualitative conclusions above may sound somewhat obvious, at least to microwave remote sensing experts. But the quantitative estimates are unprecedented to our knowledge, especially in light of the evaluation of the full uncertainty covariance matrix. One may wonder how much is the contribution of covariance matrix off-diagonal terms. To evaluate it,

$T_B$ uncertainty has been also computed considering $\mathbf{Cov}(p)$ as a diagonal matrix (i.e. all uncorrelated parameters). The difference of $\sigma(T_B)$ computed considering the full uncertainty covariance matrix and a diagonal matrix is shown in Figure 8. The contribution of off-diagonal terms goes from -1.2 to 0.6 K. It mostly affects the low-frequency oxygen wing, presumably due to line-mixing parameters and their temperature dependence, with sharp gradients in the 46-52 and 52-54





GHz frequency ranges. It also affects the atmospheric window, presumably due to water vapor continuum parameters, with a
contribution of the order of –0.3 K to –1.0 K. This demonstrates that off-diagonal terms cannot be neglected, specially in the
uncertainty characterization of window and low opacity channels of HATPRO and MP3000-A instruments.

Finally, it shall be noted that the output of this analysis is $\mathbf{Cov}(T_B)$, i.e. the full covariance matrix of $T_B$ uncertainties. A
graphical representation of $\mathbf{Cov}(T_B)$ is given in Figure 9 for HATPRO channels and U.S. standard climatology. The resulting
matrices computed for HATPRO and MP3000-A channels and the six considered climatology are provided as a
supplemental data file.

Previous studies also reported values for $\sigma(T_B)$ (Hewison et al., 2006; Hewison, 2007) and $\mathbf{Cov}(T_B)$ (Hewison 2006b),
though these were estimated from relative $T_B$ differences computed with a set of absorption models available at that time.
With respect to these values, we report (*i*) smaller uncertainty at 20-30 GHz channels, due to improved accuracy of 22 GHz
line spectroscopic parameters, and (*ii*) much larger uncertainty at 50-54 GHz channels, due to the consideration of line-
mixing parameter uncertainties, which likely cancelled out partially in the relative $T_B$ difference approach used by Hewison
(2006b; 2007).

### 5.2 Uncertainty on temperature and humidity retrievals

The uncertainty in absorption model parameters impacts the accuracy of geophysical variables retrieved from radiometric
observations through inversion methods based on a forward operator. Here, the forward operator is a radiative transfer model
(RTM) relying on the spectroscopic parameters to compute atmospheric absorption/emission, and thus the measurable $T_B$,
from atmospheric thermodynamical profiles. Examples of such inversion methods are described in Cimini et al. (2006) and
include simulation-based regression, artificial neural networks, and optimal estimation method (OEM). The OEM approach
is particularly suitable to investigate the uncertainty contribution of spectroscopic parameters, as it allows one to perform an
assessment of the total statistical uncertainty, as well as of the forward model parameter uncertainty (Rodgers, 2000). For
example, it has been used for a spectroscopic parameter sensitivity study for a millimeter/sub-millimiter limb sounder
instrument (Verdes et al. 2005) and to estimate the impact of forward model parameters on the temperature retrieval from a
multiple-channel Rayleigh-scatter lidar (Sica and Haefele, 2015).

Thus, let us consider the OEM formalism. Following Rodgers (2000), the total uncertainty covariance matrix of the retrieved
atmospheric profile $\hat{x}$ is:

$$\mathbf{Cov}(\hat{x}) = \mathbf{Cov}_m + \mathbf{Cov}_s + \mathbf{Cov}_p \tag{39}$$

where $\mathbf{Cov}_m$ and $\mathbf{Cov}_s$ are respectively the measurement and smoothing uncertainty covariance matrices, while $\mathbf{Cov}_p$ is the
model parameter uncertainty covariance matrix. $\mathbf{Cov}_p$ is related to $\mathbf{Cov}(p)$ through $K_p$, the Jacobian of the forward model





with respect to the parameters $p$, and the sensitivity of the inverse method to the measurements (also called the contribution function or gain matrix) $\boldsymbol{G}_m = \partial \boldsymbol{I}(\boldsymbol{m})/\partial \boldsymbol{m}$ as:

$$\mathbf{Cov}_p = \left(\boldsymbol{G}_m \boldsymbol{K}_p\right) \mathbf{Cov}(p) \left(\boldsymbol{G}_m \boldsymbol{K}_p\right)^{\top} \tag{40}$$


Assuming a linear Gaussian case, as usual for ground-based radiometric retrievals of atmospheric temperature and humidity profiles (Löhnert et al., 2004; Cimini et al., 2006; Hewison, 2007; Cimini et al., 2010), and calling $\mathbf{Cov}(\epsilon)$ and $\mathbf{Cov}(x_a)$ the covariance matrices of measurement and *a priori* background uncertainty, the gain matrix is given by (Rodgers, 2000):

$\quad \boldsymbol{G}_m = (\boldsymbol{K}_x^{\top} \mathbf{Cov}(\epsilon)^{-1} \boldsymbol{K}_x + \mathbf{Cov}(x_a)^{-1})^{-1} \boldsymbol{K}_x^{\top} \mathbf{Cov}(\epsilon)^{-1}$ $\hfill (41)$

where $\boldsymbol{K}_x$ is the Jacobian of the forward model with respect to the atmospheric state $x$. Finally, considering $\mathrm{T_B}$ as the measurements and recalling Eq. (18), the model parameter uncertainty covariance matrix in Eq. (40) becomes:

$\quad \mathbf{Cov}_p = \boldsymbol{G}_m \mathbf{Cov}(T_B) \boldsymbol{G}_m^{\top}$ $\hfill (42)$

which contributes to the total profiling uncertainty as in Eq.(39). Note that $\mathbf{Cov}(T_B)$ is the full spectroscopic parameter uncertainty covariance matrix estimated in Section 5.1. Accordingly, the combined uncertainty due to $O_2$ and $H_2O$ absorption model parameter is thus propagated into the retrieval space.

As an example of the spectroscopic contribution to profiling uncertainty we apply the approach described above to HATPRO channels (as in Table 4), specifically: (i) seven K-band channels (22.24 to 31.40 GHz) and (ii) seven V-band channels (51.26 to 58.0 GHz) to compute the impact on, respectively, specific humidity and temperature profile retrievals. For the sake of result reproducibility, simple diagonal $\mathbf{Cov}(\epsilon)$ and $\mathbf{Cov}(x_a)$ matrices are assumed here, with reasonable values resembling typical matrices adopted in ground-based microwave profiling (Martinet et al., 2015; Martinet et al., 2017). Specifically, we

assume a constant uncertainty for $\mathrm{T_B}$ measurements ($\mathbf{Cov}(\epsilon) = \sigma_{TB}^2 \boldsymbol{I}$, with $\sigma_{TB}$=0.5 K) and *a priori* temperature profile ($\mathbf{Cov}(x_a) = \sigma_T^2 \boldsymbol{I}$, $\sigma_T$=1.5 K), while a decreasing-with-height uncertainty for *a priori* specific humidity profile ($\sigma_Q \approx \sigma_Q(0)e^{-z/H}$, where $z$ is height in km, $\sigma_Q(0)$=3.2 g/kg, and $H$=4 km). The *a priori* background $x_a$ as well as Jacobian $\boldsymbol{K}_x$ are defined on 101 pressure levels, from 0.005 to 1050 hPa. These levels are selected to be denser close to surface (34 levels below 2 km), specifically for downwelling radiative transfer calculations. The vertical spacing of the adopted levels is

given in De Angelis et al. (2016).

The square-root of $\mathbf{Cov}_p$ diagonal terms are shown in Figures 10 and 11 for temperature and specific humidity profiling, respectively. Note that these uncertainty profiles should be considered just as indicative, as depend upon the vertical grid spacing and the choice of $\mathbf{Cov}(\epsilon)$ and $\mathbf{Cov}(x_a)$. Nonetheless, Figures 10 and 11 show that the contribution of absorption





model uncertainty to the profile retrieval uncertainty is generally not negligible. For temperature, absorption model contributes less near the surface and more in the upper atmosphere; these are respectively direct consequences of negligible uncertainty for $O_2$ opaque channels (55-58 GHz) and significant uncertainty for $O_2$ transparent channels (50-55 GHz). Above 3 km, the impact increases for colder and dryer conditions. Though less clearly, this holds also below 3 km for all but Tropical conditions, which shows a peak around 2 km. This is due to the fact that lower V-band channels (51-52 GHz) gain sensitivity to boundary layer temperature as moisture increases. These channels are the most affected by absorption model

uncertainty (Figure 6 and Table 4) and thus contribute to larger temperature uncertainty in the lower layers. For specific humidity, the absorption model contribution to uncertainty simply increases with increasing moisture. This is a direct consequence of increasing K-band $T_B$ uncertainty corresponding to increasing moisture, as seen in Figure 6.

Note that in the actual retrieval process, the contribution of absorption model parameter uncertainty to the total profiling uncertainty can be equivalently treated as $\mathbf{Cov}_p$ or as adding an absorption model term to the measurement uncertainty, i.e.

$\mathbf{Cov}(\epsilon) + \mathbf{K}_p \mathbf{Cov}(p) \mathbf{K}_p^{\mathsf{T}}$ (Rodgers, 2000).

## 6 Summary and conclusions

Radiative transfer models have general implications for atmospheric sciences, including meteorology and climate studies. Atmospheric absorption modeling is a key component of radiative transfer codes, which are extensively used for the retrieval

of atmospheric variables and the assimilation of radiometric observations into NWP. Uncertainties in atmospheric absorption models thus contribute to the uncertainty of atmospheric retrievals and observations vs. background comparison. The analysis above shows a viable approach to quantify the uncertainties of atmospheric absorption modeling and the impact on radiative transfer calculations and atmospheric retrievals. The approach relies on the estimation of the full covariance matrix of parameter uncertainties, which is necessary to compute the uncertainty of calculated $T_B$ at any given frequency. The

approach is general, and not limited to any particular instrument, technique, or frequency range. The approach can be applied to any absorption model and it can be easily extended to other frequencies and observation geometry (e.g. from satellite). To demonstrate its use quantitatively, we apply this approach to a widely-used microwave absorption model (R17, Rosenkranz 2017), focusing on the 20-60 GHz frequency range, commonly exploited for atmospheric remote sounding by ground-based MWR profilers.

We have summarized the modifications made in the last twenty years to a reference absorption model (Rosenkranz, 1998), leading to the current version of the model R17. We reviewed the spectroscopic literature searching for uncertainty estimates affecting the spectroscopic parameters entering in the absorption model code. In the considered frequency range, atmospheric absorption is dominated by water vapor and oxygen. The associated parameters and their uncertainties are reported in Tables 1 and 2, respectively for water vapor and oxygen absorption. We performed a sensitivity analysis by

perturbing each parameter by its estimated uncertainty and quantifying the impact on simulated $T_B$ for six climatology



conditions. The uncertainty of the following parameters is found to impact 20-60 GHz $T_B$ calculations by more than 0.1 K in any of the considered climatology. Concerning water vapor absorption, these are: self and foreign continuum absorption coefficients, line broadening by dry air, line intensity, temperature-dependence exponent for foreign continuum absorption, and line shift-to-broadening ratio. Concerning oxygen absorption, the dominating parameters are: line intensity, line

broadening by dry air, line mixing, temperature-dependence exponent for broadening, zero-frequency line broadening in air, temperature-dependence coefficient for line mixing. Thus, from the initial set of 319 considered parameters, 111 are retained for further analysis (6 for water vapor and 105 for oxygen). For the retained parameters, we estimated the full uncertainty covariance matrix, i.e. including parameter uncertainty variances and cross-covariance between uncertainty of different parameters. Since the spectroscopic literature provides at most the uncertainties of individual parameters, but not the

covariance between them, the off-diagonal terms of the uncertainty covariance matrix had to be estimated investigating the possible correlation between the methods used to retrieve the parameter values. The full uncertainty covariance matrix (111x111) as estimated is provided as supplemental online material.

Then, the contribution of the spectroscopic parameter uncertainty, including the covariance between them, to the uncertainty of simulated downwelling 20-60 GHz $T_B$ is calculated for six climatology conditions using the estimated uncertainty

covariance matrix (Figure 6). Dividing the 20-60 GHz range in four parts, typical $T_B$ uncertainty are: ~0.3 K (sub-Arctic winter) to nearly 1.0 K (tropical) at 20-26 GHz; ~0.4 K to 1.2 K, with additional ~0.2 K uncertainty in tropical conditions, at 26-45 GHz; up to 3.4 K inversely proportional to temperature at 45-54 GHz; below 0.5 K at 54-55 GHz, rapidly approaching zero for frequencies above 55 GHz. The dominant uncertainty contributions are water vapor line width and shift at 20-26 GHz, water vapor continuum at 26-45 GHz, oxygen line-mixing at 45-55 GHz; finally, absorption model uncertainty

becomes negligible at 55-60 GHz. Despite these qualitative conclusions may sound obvious, at least to microwave remote sensing experts, the quantitative estimates are unprecedented to our knowledge, especially in light of the evaluation of the full uncertainty covariance matrix. It is shown that off-diagonal terms affect the low-frequency oxygen wing, presumably due to covariance of line-mixing parameters and their temperature dependence, but also the atmospheric window, presumably due to covariance of water vapor continuum parameters. The total contribution depends upon frequency and

ranges from -1.2 to 0.6 K, demonstrating that off-diagonal terms cannot be neglected, especially in the uncertainty characterization of window and low opacity channels.

The resulting uncertainty on simulated $T_B$ is also calculated at the channels of two of the most common commercial MWR, i.e. HATPRO and MP3000-A. The computed $\mathbf{Cov}(T_B)$, of which one example is shown in Figure 9, are provided for the two instruments and for the six climatology conditions as supplemental online material. These matrices may directly be exploited

as the additional observation uncertainty related to absorption model in any retrieval and data assimilation procedure exploiting either of the two instruments. Just to give an example, the absorption model uncertainty is propagated to ground-based MWR retrievals, showing its impact on retrieved temperature and humidity profiles for the six climatology conditions (Figures 10 and 11). It is shown that the contribution of absorption model uncertainty to the profile retrieval uncertainty depends on climatology (increasing temperature uncertainty with decreasing average temperature, increasing humidity




uncertainty with increasing moisture), and it is generally not negligible, though the actual values depend on retrieval settings (such as a priori information, vertical spacing, among others).

Finally, let us underline that the presented uncertainty quantification contributes to a better understanding of the total uncertainty affecting radiometric products, thus reducing the chances of systematic errors in NWP data assimilation and observations-derived climate trends. Future work may include the application of the proposed approach to higher frequencies

and upwelling $T_B$, requiring a new sensitivity analysis, as well as further modification to the R17 absorption model to account for recent findings from spectroscopic laboratory experiments (e.g. inter-branch coupling suggested by Makarov et al. (2013), temperature exponent $n_a$ suggested by Koshelev et al. (2016), consideration of speed dependence of collisional relaxation effect, influencing diagnostic line profiles as shown in Koshelev et al. (2018)).

**Appendix A: Modifications to R98 leading to R17**

The following two sections review the set of modifications to the R98 model, respectively for water vapor and oxygen absorption, proposed in the open literature in the last 20 years and subsequently imported in the current R17 version of the model.

**A.1 Water vapor**

The R98 model uses 15 water vapor lines, similar to the strongest lines used in MPM89, while the other 15 lines have been omitted as they were judged to have negligible impact. For the water vapor continuum absorption, the model combines the foreign-broadened component from MPM87 with the self-broadened from MPM93, increased by 15% and 3%, respectively, to compensate for the line truncation at cut-off frequency (±750 GHz). This model is still maintained and there have been

several modifications since the 1998 version.

Since 2003, the model includes the pressure line shift mechanism investigated by Tretyakov et al. (2003) and Golubiatnikov et al. (2005). For the 22.23 and 183.31 GHz absorption lines, the only two relevant for the frequency range under study here, the main modifications are the adoption of the air-broadened line widths determined in Payne et al. (2008) using ground-based radiometric measurements, leading to -5.1% and +4.5% line-width change, respectively. The -5% modification to the

22.23 GHz line width was already proposed by the independent investigation of Liljergren et al. (2005). Other modifications for the 22.23 and 183.31 GHz absorption lines are in line intensity (+0.3% and +0.5%, i.e. from HITRAN 1992 to 2012 update), in the temperature exponent of air-broadening (+10% and +20%, respectively), in the self-broadened line-width (+0.8% and -1.0%), while the temperature exponent of self-broadening only changed for the 22.23 GHz line (+64%).

Parameters for higher frequency lines (321-916 GHz) were modified according to different sets of spectroscopic

measurements (Colmont et al., 1999; Podobedov et al., 2004; Koshelev et al., 2007; Golubiatnikov et al., 2008; Koshelev,



2011; Tretyakov et al., 2013), leading to modifications in air-broadened line-width (order of 1-15%), temperature exponent of air-broadening (2-5%), and self-broadened line-width (1-9%). Other line parameters are from HITRAN 2012 database (Rothman et al., 2013).

Concerning the water vapor continuum, the main modifications follow the results of Turner et al. (2009) suggested by an

analysis of ground-based observations at 150 GHz. The suggested adjustments to the two components of the water vapor continuum in R98 model are in opposite directions (i.e., increasing the contribution from the foreign-broadened component while decreasing contribution from the self-broadened component). Figure 12 plots $C_s$ vs. $C_f$ for the R98 model and its modification by Turner *et al*. (2009) with their respective uncertainty contours. These uncertainties are conditioned on the nominal values of $n_{cs}$ and $n_{cf}$, which are the same in both models. The uncertainty ellipse for Turner *et al*. is drawn using the

correlation coefficient of -0.87 found in Section 4.1.3. Note that the details of continuum and resonant absorption are inextricably related in any model, meaning that the empirical definition of the continuum (Eq. (8)) implies that the parameters must be used only with exactly the same resonance absorption they were defined with. Thus, the adjustment factors have been recomputed in 2015 accounting for the resonant line adjustments discussed above, leading to +9.8% and -21.1% change from R98 in air-broadened and self-broadened coefficients, respectively. The results of Turner et al. (2009)

are indirectly supported by the analysis of Payne et al. (2011). In fact, Payne et al. (2011) developed adjustment factors for the MT_CKD water vapor continuum model (Clough et al., 2005; Mlawer et al., 2012), which agree within the stated error bars with those given in Turner et al. (2009) for the same MT_CKD model.

More recently, two papers presented further modifications to the spectroscopy underlying microwave remote sensing of atmospheric water vapor, i.e. Tretyakov (2016) and Koshelev et al. (2018). Tretyakov (2016) presents a historic review,

discussing in chronological order the measurement and analysis that lead to estimates of spectroscopic parameters for the water vapor absorption continuum and resonant lines near 22 and 183 GHz. Tretyakov (2016) also provides an expert assessment for the best estimate of the spectroscopic parameter values and their uncertainty, based on the analysis of all the available data. These parameter values provide the best fit of the absorption model to the available data, taking into account the measurement errors reported by the authors and the probabilities of possible systematic errors. In almost all cases, with

the exception of the 22-GHz line self-broadening, the estimated parameter values agree within uncertainty limits with those given in HITRAN, though in most cases HITRAN uncertainty estimates are more conservative. Concerning the water vapor continuum absorption, Tretyakov (2016) finds that the adjustments to R98 proposed by Turner *et al*. (2009), based on zenith-looking ground-based radiometric observation, lead to a worse fit to the laboratory and field (parallel to Earth-surface path) measurements, particularly noticeable in the self component. However, Figure 12 shows that the model uncertainties have

appreciable overlap. Finally, Koshelev et al. (2018) presents laboratory measurements devoted to refining of 22 GHz line shape parameters. Koshelev et al. (2018) suggest line width values within the uncertainty of those given by Tretyakov (2016), though with smaller estimated uncertainty by a factor of ~3 (air-broadening) and ~10 (self-broadening). Similarly, the air-broadening shift parameter agrees with that of Tretyakov (2016) with an estimated uncertainty reduced by a factor of




~3. Conversely, the uncertainty of the self-broadening shift parameter is reduced by a factor ~1.5, and the values from
Tretyakov (2016) and Koshelev et al. (2018) do not fit within the stated uncertainty.

### A.2 Oxygen

The R98 model adopts the same oxygen line parameters as given in MPM92, except for submillimeter frequencies, where
frequency and intensity are taken from the HITRAN 1992 database (Rothman et al., 1992). Other differences with respect to
MPM92 are: the temperature dependence (1/T) for 118.75 GHz line width; the temperature dependence of submillimetre line
widths being equal to that of lines in the 60 GHz band (e.g. $1/T^{n_a}$, with $n_a$=0.8). Concerning the line mixing model, MPM as
well as R98 model exploit first-order mixing with coefficients derived by the method given in Rosenkranz (1988). The
following modifications have been implemented in R17.

The line intensities are from the HITRAN 2004 database (Rothman et al., 2005). The zero-frequency line intensity is from
the Jet Propulsion Laboratory (JPL) catalog (https://spec.jpl.nasa.gov/, Pickett et al., 1998). The line central frequencies and
width coefficients for the 60-GHz band are taken from Tretyakov et al. (2005), who report measurements for precise
broadening and central frequencies of fine-structure lines and a revision of line mixing coefficients. The effect of different
values for the 60-GHz line parameters on MWR simulations and retrievals was shown to be significant both for ground-
based (Cadeddu et al., 2007) and satellite (Boukabara et al., 2005) observations. In particular, Cadeddu et al. (2007) show
that the parameter values proposed by Tretyakov et al. (2005) lead to better agreement with two independent datasets of
ground-based MWR observations than do those found in HITRAN (Rothman et al., 2005; Hoke et al. 1989), and also that
these modifications are essential to reduce the clear-sky bias in the liquid-water path retrievals.

The line-width and line-mixing coefficients for the 118 GHz line are taken from Tretyakov et al. (2004)., who report results
of laboratory investigations of the pressure-dependent parameters of the single 118-GHz line. The submillimetre line-widths
are from Golubiatnikov and Krupnov (2003), except the one at 234 GHz line that comes from Drouin (2007).

Makarov et al. (2011) proposed a model for the 60-GHz absorption band based on the second-order line-mixing expansion of
Smith (1981), showing improved fit of observed absorption profiles between 54 and 65 GHz, but this model is not adopted in
R17. In fact, during this analysis, significant absorption differences (~10%) were found in the band wings (e.g. ~50-53 GHz)
comparing calculations made with Makarov et al. (2011) line-mixing coefficients against original measurements from Liebe
et al. (1992). This was attributed to systematic errors in $O_2$ concentration of the order of 0.5-1.5% in the 245-335 K
temperature range. Makarov and colleagues are currently working on a revised second-order model (personal
communication, 2018).

For the dry continuum, R98 only considered the $N_2$-$N_2$ contribution with a pure $\nu^2$ dependence. This is a particular case of
Eq. (7) and (15), with $\varepsilon(\nu, T) = 0$ and $f(\nu) = 1$. This was revised (Rosenkranz et al., 2006) fitting $f(\nu)$ as Eq.(16), through
the data of Borysow and Frommhold (1986) and including the $N_2$-$O_2$ and $O_2$-$O_2$ bimolecular absorption with a constant value
for $\varepsilon$ suggested by Pardo et al. (2001) and later by Boissoles et al. (2003). The latter is used in R17.





In order to consider the broadening of oxygen lines by water vapor with little modifications to the original model, R17 adopts the mean value of the water-to-air broadening ratio suggested by Koshelev et al. (2015).

More recently, Koshelev et al. (2016) report measurements of line widths and their temperature exponents for twelve oxygen

lines (rotational quantum number N ranging from 1 to 19). The fixed value of the temperature exponent ($n_a$=0.8) adopted in the MPM and R98/R17 models fits the value reported in Makarov et al. (2008) for the 1- line (0.785(35)) but falls outside the mean value (0.765(11)) reported by Koshelev et al. (2016). This suggests that the temperature exponent values suggested by Koshelev et al. (2016), or their mean value, could be adopted to increase the accuracy of absorption modelling.

**Acknowledgements**

This work was partially supported by the EU H2020 project GAIA-CLIM (Ares(2014)3708963/Project 640276). M. Tretyakov and M. Koshelev acknowledge the State project No 0035-2014-009. D. Cimini acknowledges the useful advices from Stefan Bühler, Richard Larsson, Oliver Lemke in the early stage of the analysis.

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



| Symbol [units] | Parameter | Value | Uncertainty | Reference |
|---|---|---|---|---|
| $\nu_i$ [kHz] | Resonant line frequency at 22 GHz at 183 GHz | 22235079.85 183310087 | 0.05 1 | Kukolich 1969 Golubiatnikov et al. 2006 |
| $S_i$ [Hz*cm$^2$] | Resonant line intensity at 22 GHz at 183 GHz | $1.3161 \cdot 10^{-14}$ $2.3222 \cdot 10^{-12}$ | 1% 1% | Polyansky et al., 2018 Tretyakov 2016 |
| $n_S$ [unitless] | Resonant line intensity temperature dependence exponent | 2.5 | 0.5% | Gamache et al., 2017 This work |
| $E_{low}$ [cm$^{-1}$] | Resonant line lower-state energy at 22 GHz at 183 GHz | 446.5106590 136.163927 | $4 \cdot 10^{-8}$ % $7 \cdot 10^{-7}$ % | Tennyson et al., 2013 |
| $\gamma_a$ [GHz/bar] | Resonant line air-broadening at 22 GHz at 183 GHz | 2.688 2.945 | 0.039 0.015 | Koshelev et al. 2018 Tretyakov, 2016 |
| $\gamma_w$ [GHz/bar] | Resonant line water-broadening at 22 GHz at 183 GHz | 13.281 14.77 | 0.039 0.37 | Koshelev et al. 2018 Tretyakov, 2016 |
| $n_a$ [unitless] | Resonant line air-broadening temperature dependence exponent at 22 GHz at 183 GHz | 0.70 0.74 | 0.05 0.03 | Payne et al. 2008 Tretyakov 2016 |
| $n_w$ [unitless] | Resonant line water-broadening temperature dependence exponent at 22 GHz at 183 GHz | 1.20 0.78 | 0.5 0.08 | Cazzoli et al. 2007 Bauer et al. 1989 Tretyakov 2016 |
| R [unitless] | Resonant line shift to broadening ratio at 22 GHz at 183 GHz | -0.0089 -0.0245 | 0.0106 0.0026 | Koshelev et al. 2018 Tretyakov, 2016 |
| $C_f$ [km$^{-1}$ mb$^{-2}$ GHz$^{-2}$] | Foreign-broadened continuum | $5.96 \cdot 10^{-10}$ | $5.5 \cdot 10^{-11}$ | Rosenkranz 1998 Turner et al. 2009 |
| $C_s$ [km$^{-1}$ mb$^{-2}$ GHz$^{-2}$] | Self-broadened continuum | $1.42 \cdot 10^{-8}$ | $3.2 \cdot 10^{-9}$ | Rosenkranz 1998 Turner et al. 2009 |
| $n_{cf}$ [unitless] | Foreign-broadened continuum temperature dependence exponent | 0.0 | 0.8 | Rosenkranz 1998 Tretyakov, 2016 Koshelev et al. 2011 |
| $n_{cs}$ [unitless] | Self-broadened continuum temperature dependence exponent | 4.5 | 0.6 | Rosenkranz 1998 Tretyakov, 2016 Koshelev et al. 2011 |

Table 1: List of water-vapor parameters perturbed in the sensitivity analysis.





210

| Symbol [units] | Parameter | Value | Uncertainty | Reference |
|---|---|---|---|---|
| $\nu_i$ [kHz] | Resonant line frequency | Table 1* | Table 1* | Tretyakov et al., 2005 |
| $S_i$ [Hz/cm$^2$] | Resonant line intensity | HITRAN 2004 | 1% | Rothman et al., 2005 and this work |
| $n_S$ [unitless] | Resonant line intensity temperature dependence exponent | 2.0 | 0.1% | Gamache et al., 2017 This work |
| $E_{low}$ [cm$^{-1}$] | Resonant line lower-state energy | HITRAN 2004 | 0.25% | This work |
| $\gamma_i$ [GHz/bar] | Resonant line air-broadening | Table 5* | Table 1* + this work | Tretyakov et al., 2005 Koshelev et al. 2016 |
| $n_a$ [unitless] | Resonant line air-broadening temperature dependence exponent | 0.80 | 0.05 | Koshelev et al. 2016 |
| $Y_i$ | Resonant line mixing | Table 5* | This work | Tretyakov et al., 2005 |
| $V_i$ | Resonant line mixing temperature dependence | Table 5* | This work | Liebe et al., 1992 Tretyakov et al., 2005 |
| $r_{w2a}$ [unitless] | Resonant line water-to-air broadening ratio | 1.20 | 0.05 | Koshelev et al. 2015 |
| $\gamma_0$ [GHz/bar] | Zero-frequency line pressure broadening | 0.56 | 0.05 | This work (based on Danese & Partrige, 1989) |

Table 2: List of oxygen parameters perturbed in the sensitivity analysis. *Table 1 and 5 from Tretyakov et al., 2005.





**A. Covariance matrix**

| | $C_f(300)$ | $C_s(300)$ | $\gamma_a(296)$ | $S(296)$ | $n_{cf}$ | $R$ |
|---|---|---|---|---|---|---|
| $C_f(300)$ | $4.58\cdot10^{-21}$ | $-1.57\cdot10^{-19}$ | $-2.63\cdot10^{-15}$ | $-5.86\cdot10^{-30}$ | $-3.08\cdot10^{-11}$ | $-7.86\cdot10^{-18}$ |
| $C_s(300)$ | $-1.57\cdot10^{-19}$ | $1.05\cdot10^{-17}$ | $0$ | $-2.31\cdot10^{-29}$ | $0$ | $0$ |
| $\gamma_a(296)$ | $-2.63\cdot10^{-15}$ | $0$ | $1.52\cdot10^{-3}$ | $0$ | $0$ | $5.05\cdot10^{-6}$ |
| $S(296)$ | $-5.86\cdot10^{-30}$ | $-2.31\cdot10^{-29}$ | $0$ | $1.66\cdot10^{-32}$ | $0$ | $0$ |
| $n_{cf}$ | $-3.08\cdot10^{-11}$ | $0$ | $0$ | $0$ | $0.64$ | $0$ |
| $R$ | $-7.86\cdot10^{-18}$ | $0$ | $5.05\cdot10^{-6}$ | $0$ | $0$ | $1.12\cdot10^{-4}$ |

|215

**B. Correlation matrix**

| | $C_f(300)$ | $C_s(300)$ | $\gamma_a(296)$ | $S(296)$ | $n_{cf}$ | $R$ |
|---|---|---|---|---|---|---|
| $C_f(300)$ | $1$ | $-0.71$ | $-0.001$ | $-7\cdot10^{-4}$ | $-0.57$ | $-1\cdot10^{-5}$ |
| $C_s(300)$ | $-0.71$ | $1$ | $0$ | $-6\cdot10^{-5}$ | $0$ | $0$ |
| $\gamma_a(296)$ | $-0.001$ | $0$ | $1$ | $0$ | $0$ | $0.01$ |
| $S(296)$ | $-7\cdot10^{-4}$ | $-6\cdot10^{-5}$ | $0$ | $1$ | $0$ | $0$ |
| $n_{cf}$ | $-0.57$ | $0$ | $0$ | $0$ | $1$ | $0$ |
| $R$ | $-1\cdot10^{-5}$ | $0$ | $0.01$ | $0$ | $0$ | $1$ |

Table 3: Covariance (A) and correlation (B) matrices corresponding to spectroscopic water-vapor parameter uncertainties as derived in Section 4. Note that $C_f$ and $C_s$ are evaluated at $T_0=300$ K, while $\gamma_a$ and $S$ at $T_0=296$ K.

|220



| | 22.24 | 23.04 | 23.84 | 25.44 | 26.24 | 27.84 | 31.40 | 51.26 | 52.28 | 53.86 | 54.94 | 56.66 | 57.30 | 58.00 |
|---|---|---|---|---|---|---|---|---|---|---|---|---|---|---|
| **Trop** | 0.92 | 0.83 | 0.68 | 0.54 | 0.52 | 0.53 | 0.61 | 2.62 | 2.73 | 1.00 | 0.13 | 0.02 | 0.02 | 0.02 |
| **MidS** | 0.73 | 0.66 | 0.54 | 0.43 | 0.42 | 0.42 | 0.48 | 2.67 | 2.82 | 1.03 | 0.12 | 0.02 | 0.01 | 0.01 |
| **MidW** | 0.35 | 0.34 | 0.33 | 0.33 | 0.34 | 0.36 | 0.42 | 3.01 | 3.18 | 1.10 | 0.11 | 0.01 | 0.01 | 0.01 |
| **SubS** | 0.58 | 0.52 | 0.44 | 0.37 | 0.36 | 0.37 | 0.44 | 2.78 | 2.95 | 1.07 | 0.12 | 0.02 | 0.02 | 0.02 |
| **SubW** | 0.30 | 0.30 | 0.31 | 0.32 | 0.33 | 0.36 | 0.42 | 3.13 | 3.31 | 1.13 | 0.09 | 0.00 | 0.00 | 0.00 |
| **USstd** | 0.46 | 0.42 | 0.37 | 0.34 | 0.34 | 0.36 | 0.42 | 2.86 | 3.04 | 1.12 | 0.14 | 0.02 | 0.02 | 0.02 |

Table 4: Uncertainty on simulated $T_B$ ($\sigma(T_B)$) at 14 HATPRO channel central frequencies due to the uncertainty in $O_2$ and $H_2O$ absorption model parameters. $\sigma(T_B)$ is computed as the square root of the diagonal terms of $Cov(T_B)$, which was estimated considering the six climatological atmospheric conditions introduced in Figure 1.






| | 22.23 | 22.50 | 23.03 | 23.83 | 25.00 | 26.23 | 28.00 | 30.00 | 51.25 | 51.76 | 52.28 | 52.80 | 53.34 | 53.85 | 54.40 | 54.94 | 55.50 | 56.02 | 56.66 | 57.29 | 57.96 | 58.80 |
|---|---|---|---|---|---|---|---|---|---|---|---|---|---|---|---|---|---|---|---|---|---|---|
| **Trop** | 0.92 | 0.92 | 0.84 | 0.69 | 0.57 | 0.52 | 0.53 | 0.57 | 2.62 | 2.74 | 2.73 | 2.43 | 1.79 | 1.02 | 0.39 | 0.13 | 0.05 | 0.03 | 0.02 | 0.02 | 0.02 | 0.02 |
| **MidS** | 0.73 | 0.73 | 0.66 | 0.54 | 0.45 | 0.42 | 0.42 | 0.45 | 2.66 | 2.82 | 2.82 | 2.52 | 1.85 | 1.04 | 0.39 | 0.12 | 0.05 | 0.03 | 0.02 | 0.01 | 0.01 | 0.01 |
| **MidW** | 0.35 | 0.35 | 0.34 | 0.33 | 0.33 | 0.34 | 0.36 | 0.40 | 3.00 | 3.18 | 3.18 | 2.83 | 2.05 | 1.12 | 0.39 | 0.11 | 0.03 | 0.02 | 0.01 | 0.01 | 0.01 | 0.01 |
| **SubS** | 0.58 | 0.57 | 0.52 | 0.44 | 0.38 | 0.36 | 0.38 | 0.41 | 2.77 | 2.94 | 2.95 | 2.64 | 1.94 | 1.09 | 0.40 | 0.12 | 0.05 | 0.03 | 0.02 | 0.02 | 0.02 | 0.02 |
| **SubW** | 0.30 | 0.30 | 0.30 | 0.31 | 0.32 | 0.33 | 0.36 | 0.39 | 3.12 | 3.31 | 3.31 | 2.95 | 2.13 | 1.15 | 0.39 | 0.09 | 0.02 | 0.00 | 0.00 | 0.00 | 0.00 | 0.00 |
| **USstd** | 0.46 | 0.45 | 0.42 | 0.37 | 0.34 | 0.34 | 0.36 | 0.40 | 2.85 | 3.03 | 3.04 | 2.73 | 2.01 | 1.14 | 0.43 | 0.14 | 0.06 | 0.04 | 0.02 | 0.02 | 0.02 | 0.02 |

Table 5: As Table 4 but at 22 central frequencies of MP3000-A channels.






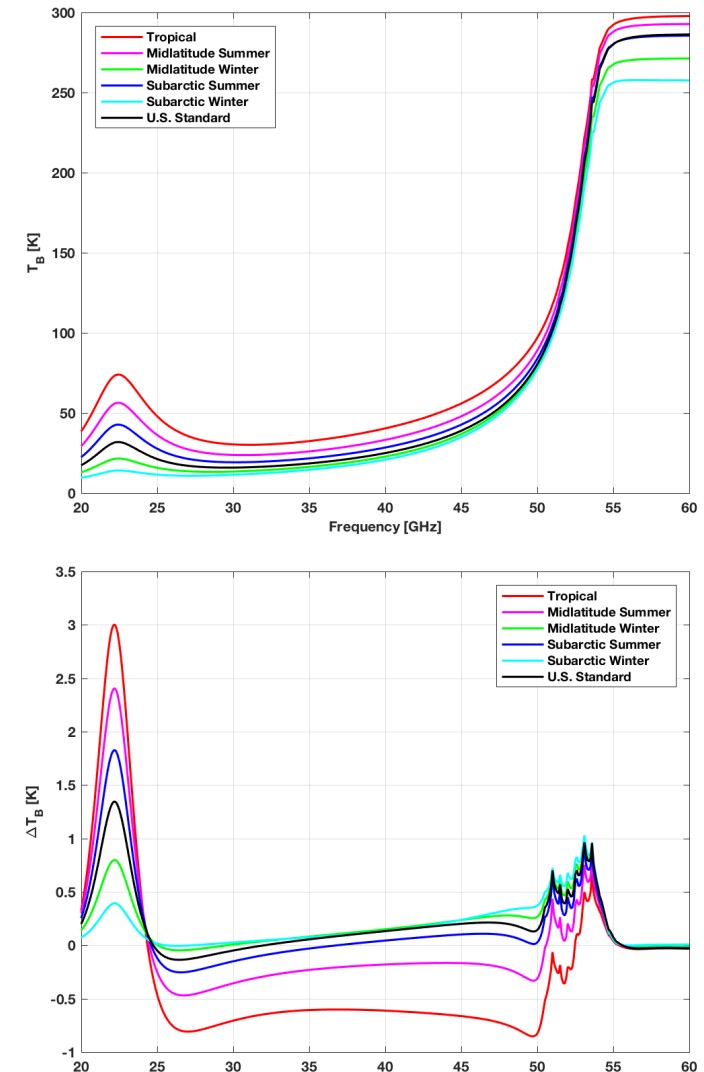

**Figure 1: (Top) Zenith downwelling $T_B$ computed using six reference atmosphere climatology conditions with the R17 model. (Bottom) Difference between $T_B$ computed with the current and reference versions (R17 minus R98) for the six atmosphere climatology conditions. Note the features at 22 GHz, mainly attributable to the updated line width (Payne et al., 2008), at 25-50 GHz, due to the scaled continuum (Turner et al., 2009), and at 50-55 GHz, related to revised coefficients for the 60 GHz band (Tretyakov et al. 2005).**





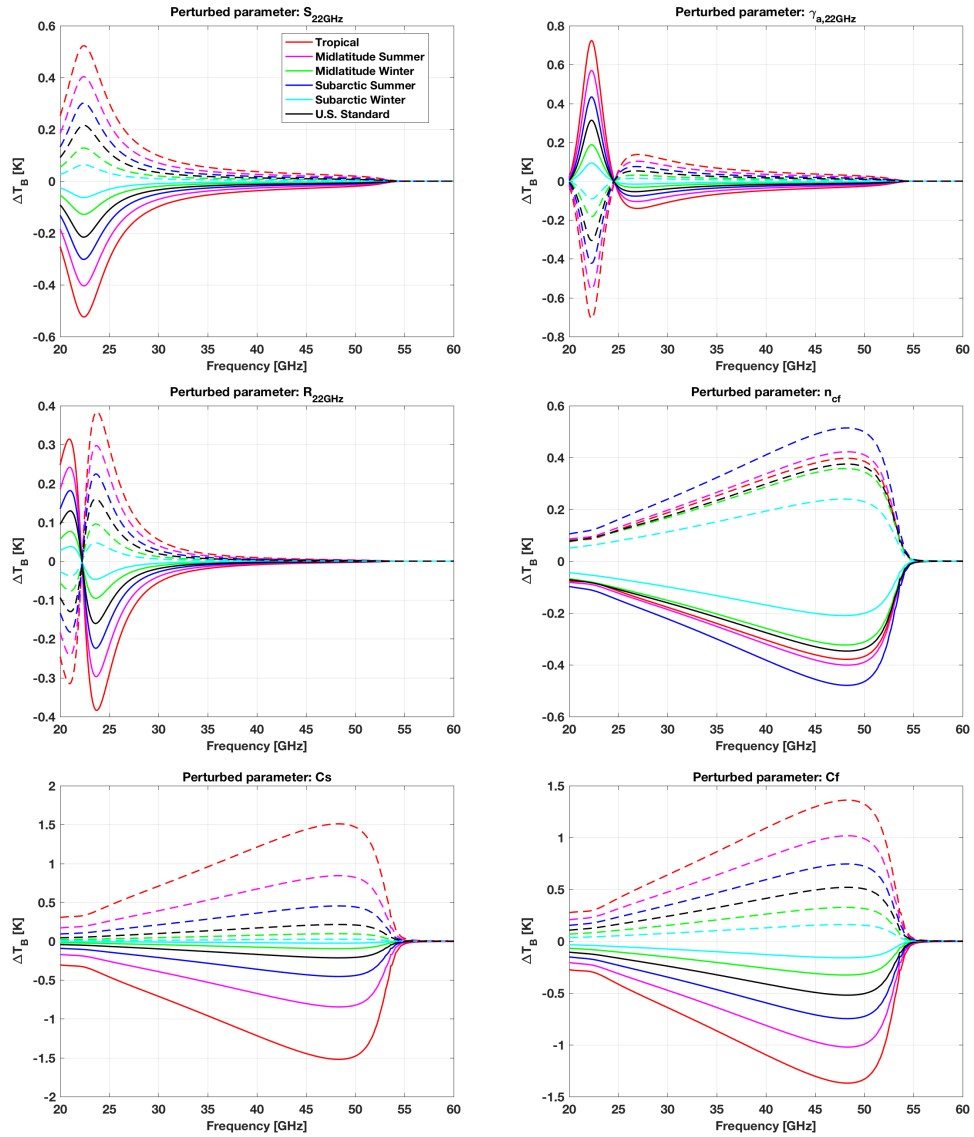

**Figure 2: Sensitivity of modelled T$_B$ to water vapor absorption parameters. Top: line intensity (S$_i$) and air-broadening (γ$_{i,a}$) at 22 GHz. Middle: Shift-to-broadening ratio (R$_i$) at 22 GHz and foreign-broadening temperature-dependence exponents (n$_{cf}$). Bottom: Self- (C$_s$) and foreign- (C$_f$) induced broadening coefficients. Solid lines correspond to negative perturbation (value − uncertainty), while dashed lines to positive perturbation (value + uncertainty).**






**Figure 3: Sensitivity of modelled T$_B$ to oxygen absorption parameters. Top: line intensity ($S_i$) and air-broadening ($\gamma_{i,a}$). Middle: air-broadening temperature-dependence exponents ($n_a$) and non-resonant pseudo-line broadening ($\gamma_{nr}$). Bottom: mixing coefficients ($y_i$) and mixing temperature-dependence coefficients ($V_i$). Note that the perturbation to $S_i$ and $n_a$ affect all lines, while**





for the other resonant line parameters we show the impact of the perturbation to just one line (N=25-) as an example. Solid lines
correspond to negative perturbation (value - uncertainty), while dashed lines to positive perturbation (value + uncertainty).

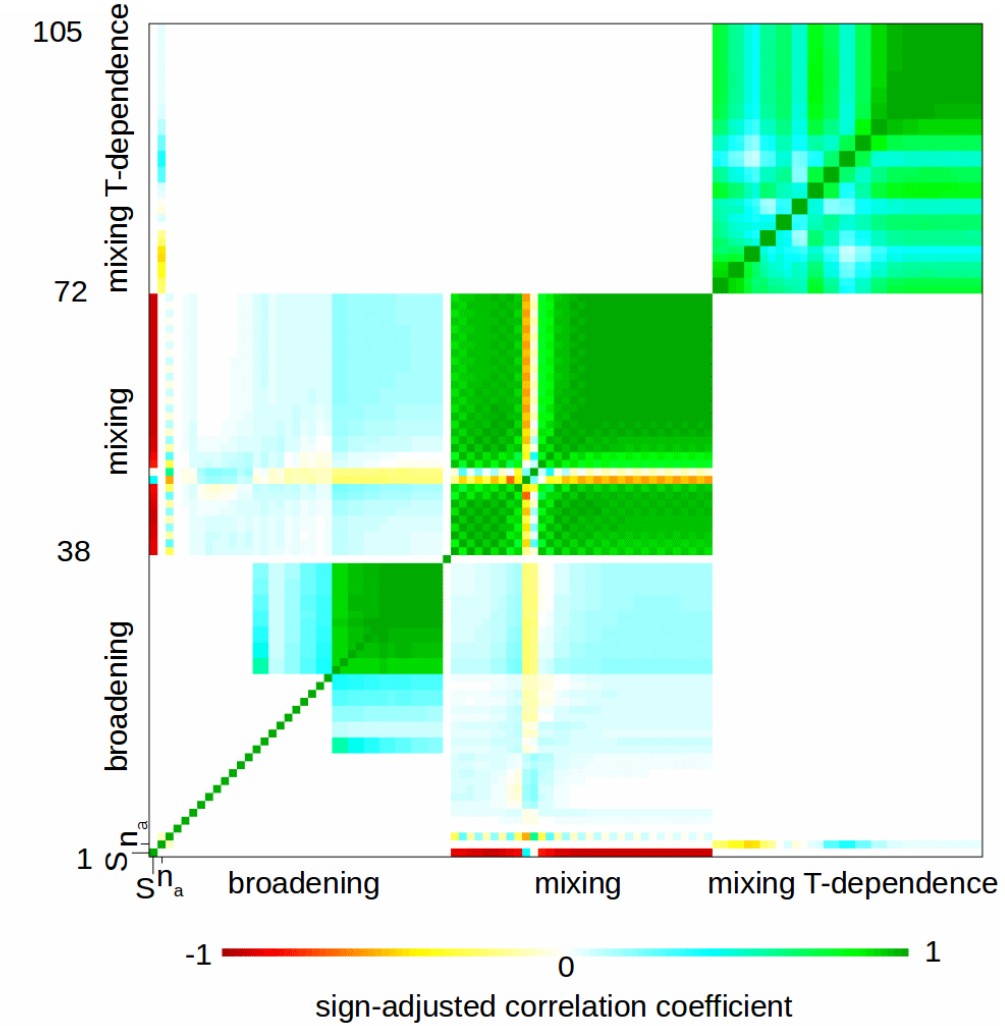

Figure 4: Uncertainty matrix for oxygen absorption as a color-scale image of sign-adjusted correlation coefficients ($\rho_{SA}$). See Eq.
(29) in Section 4.2 for the definition of $\rho_{SA}$. The y-axis's label shows selected parameter indexes. The parameters are ordered as
follows: #1) S(300), #2) $n_a$, #3) $\gamma_0$(300), #4-37) $\gamma_a$(300), #38-71) y(300), #72-105) V. The last three parameter types are ordered
following the $O_2$ rotational quantum number N=1-,1+,3-, ...33-,33+.



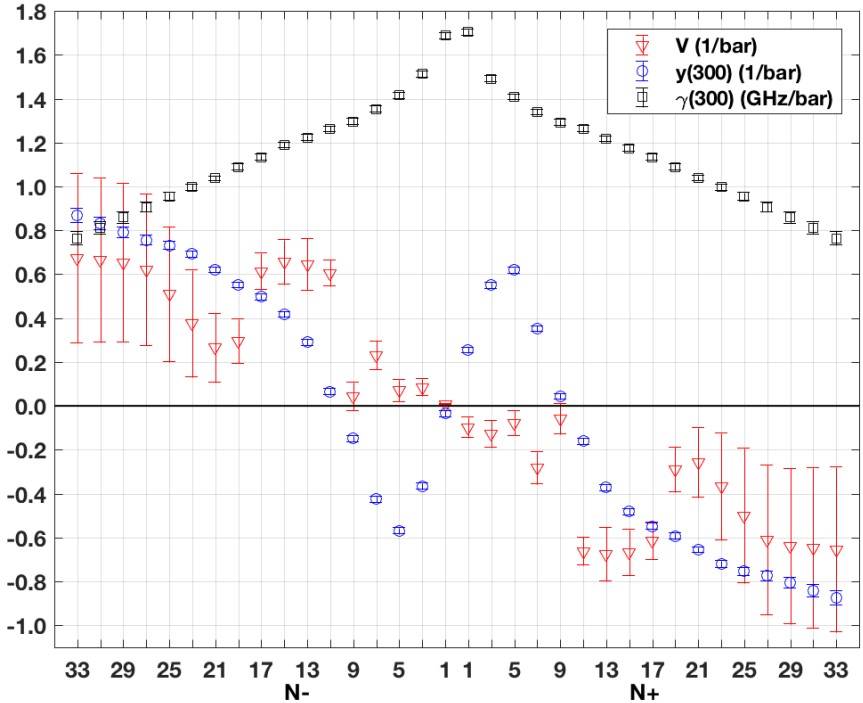

**Figure 5:** Oxygen-line parameters as a function of rotational quantum number N: line-width $\gamma_a(300)$ (squares), line-mixing y(300) (circles), and line-mixing temperature coefficients V (triangles). Error bars indicate ±1-σ uncertainties.



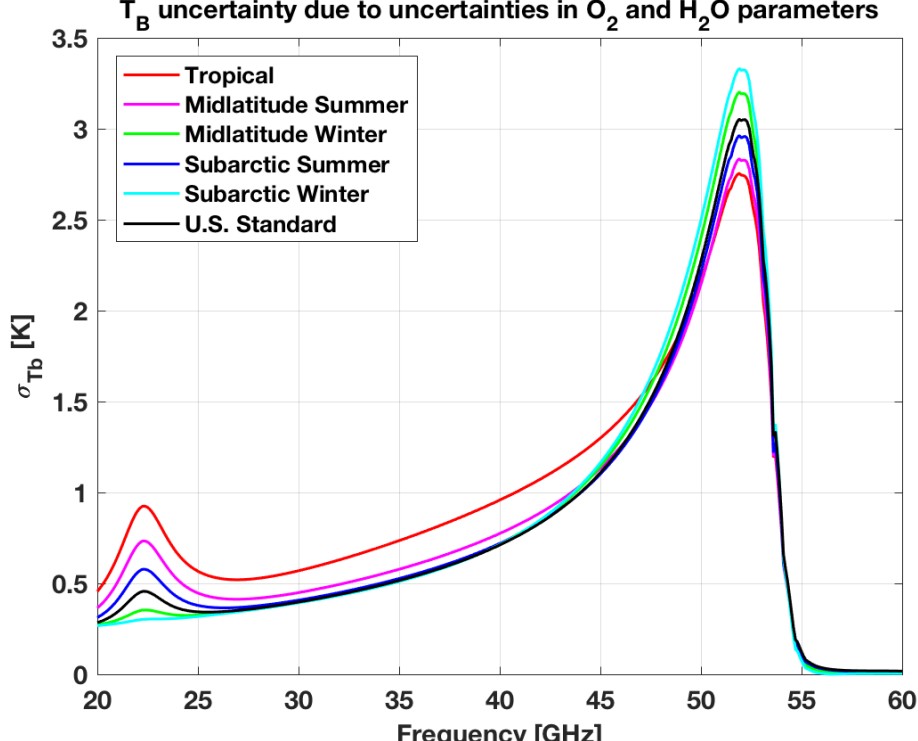

**Figure 6: Zenith downwelling $T_B$ uncertainty ($\sigma(T_B)$) due to the uncertainty in $O_2$ and $H_2O$ absorption model parameters. Six climatological atmospheric conditions (color-coded) have been used to compute $K_p$. $\sigma(T_B)$ is computed as the square root of the diagonal terms of $Cov(T_B)$.**






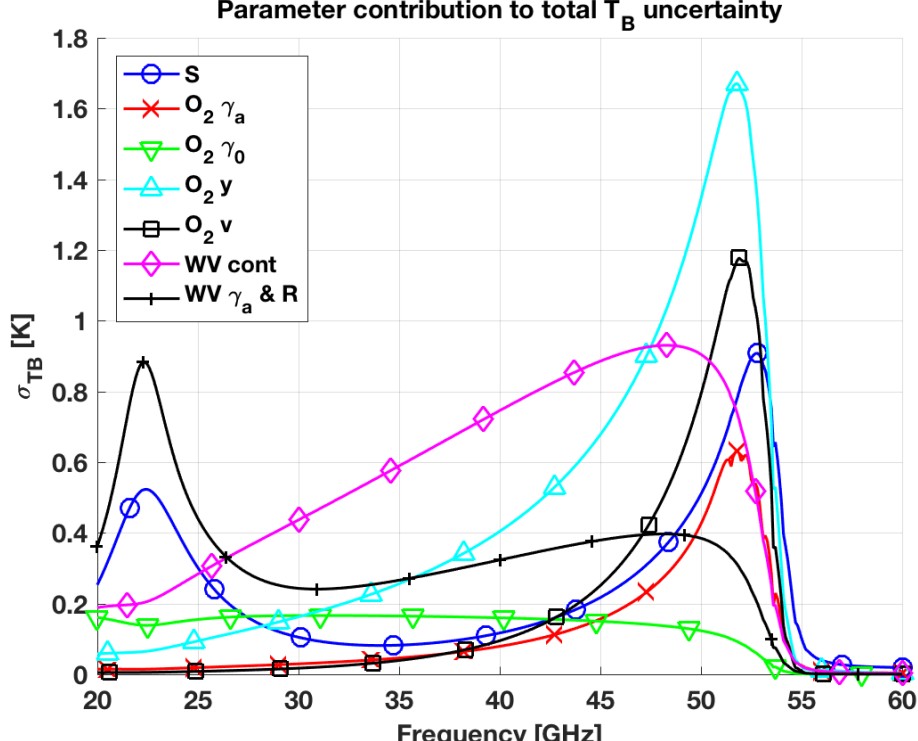

**Figure 7: Contributions to zenith downwelling $T_B$ uncertainty ($\sigma(T_B)$) due to the different types of $O_2$ and $H_2O$ absorption model parameters. Tropical climatology conditions are used here. The parameters are grouped into 7 types: intensity S (for both $O_2$ and $H_2O$), $O_2$ line width $\gamma_a$, $O_2$ zero-frequency line width $\gamma_0$, $O_2$ line mixing ($y$), $O_2$ line mixing temperature dependence ($v$), $H_2O$ continuum, $H_2O$ line width $\gamma_a$ and shift-to-width ratio $R$.**




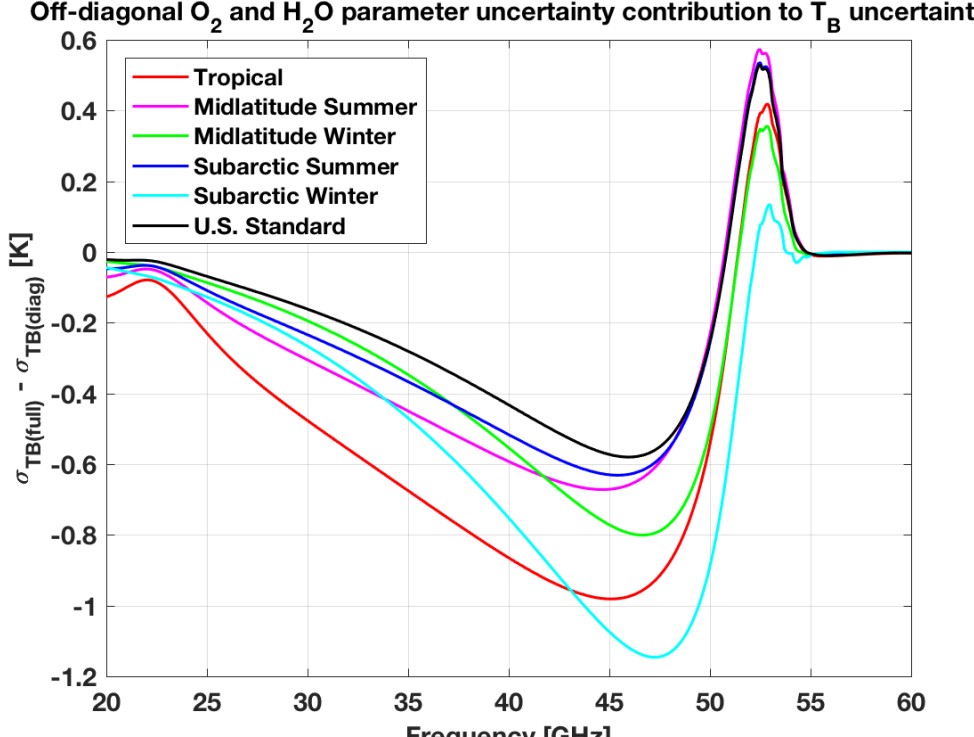

**Figure 8:** Difference between σ(T_B) as computed considering the full uncertainty covariance matrix and its diagonal matrix (i.e. off-diagonal terms are set to zero).





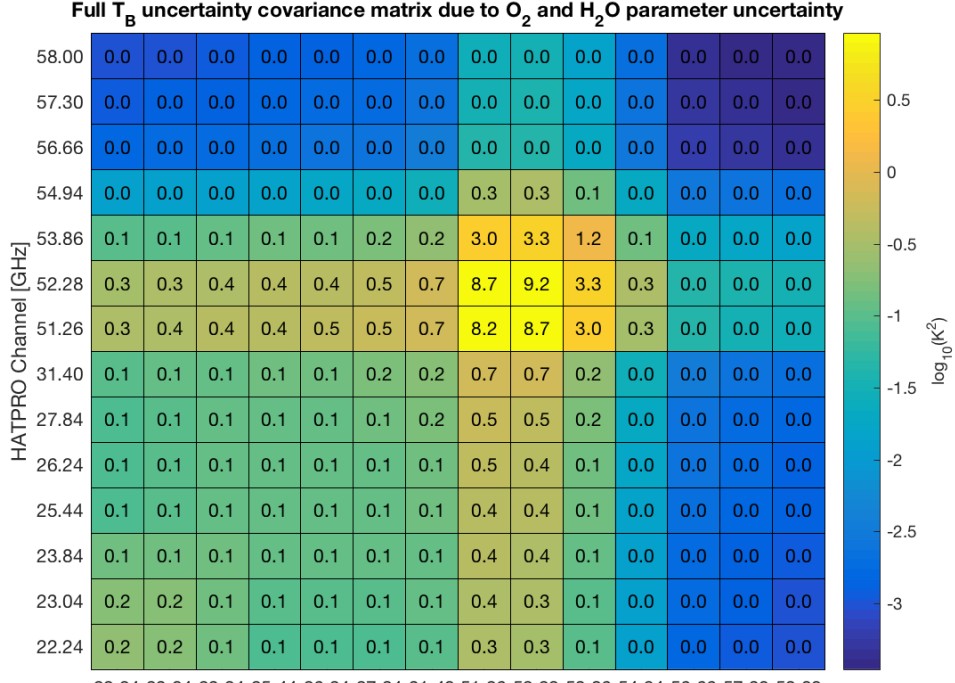

**Figure 9: $T_B$ uncertainty covariance matrix due to $O_2$ and $H_2O$ absorption model parameter uncertainty at HATPRO channels for**
**U.S. standard climatology. Numbers in the table are in $K^2$, while the colour scale is in $\log_{10}(K^2)$.**



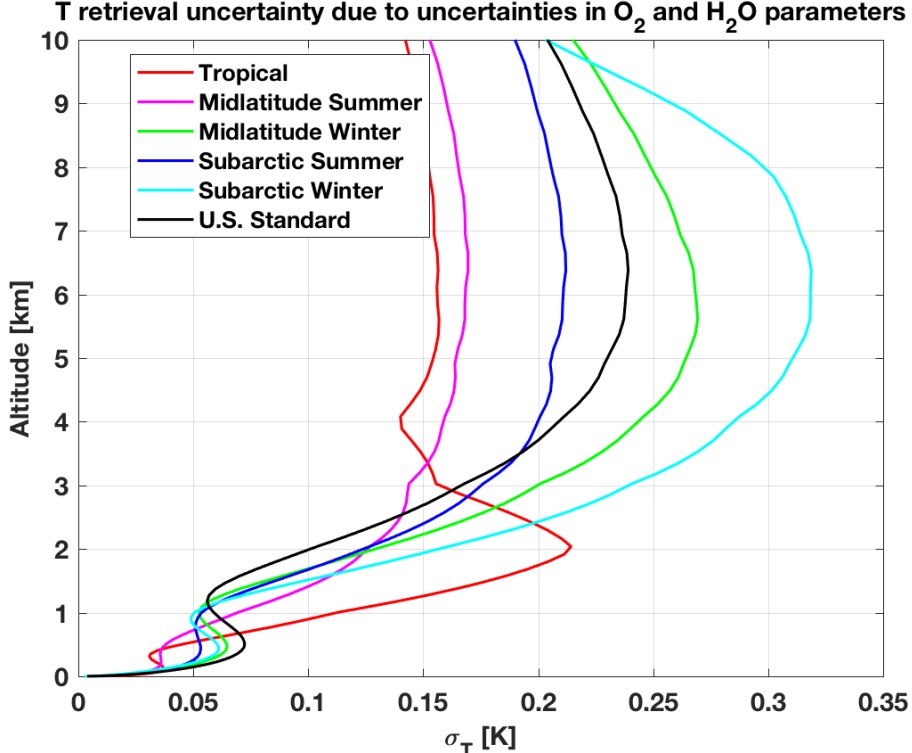

**Figure 10: Uncertainty in temperature retrievals from ground-based MWR due to the uncertainty in $O_2$ and $H_2O$ absorption model parameters. The observation vector considered here consists in $T_B$ at the 14 HATPRO channels. Six climatological atmospheric conditions (color-coded) have been used to compute $K_b$ and $K_x$. The square root of the diagonal terms of $Cov_p$ are shown. 101 pressure levels from 0.005 to 1050 hPa are used here. These levels have been selected specifically to be denser close to surface (34 levels below 2 km). The vertical spacing of levels is given in Figure 1 of De Angelis et al. (2016).**




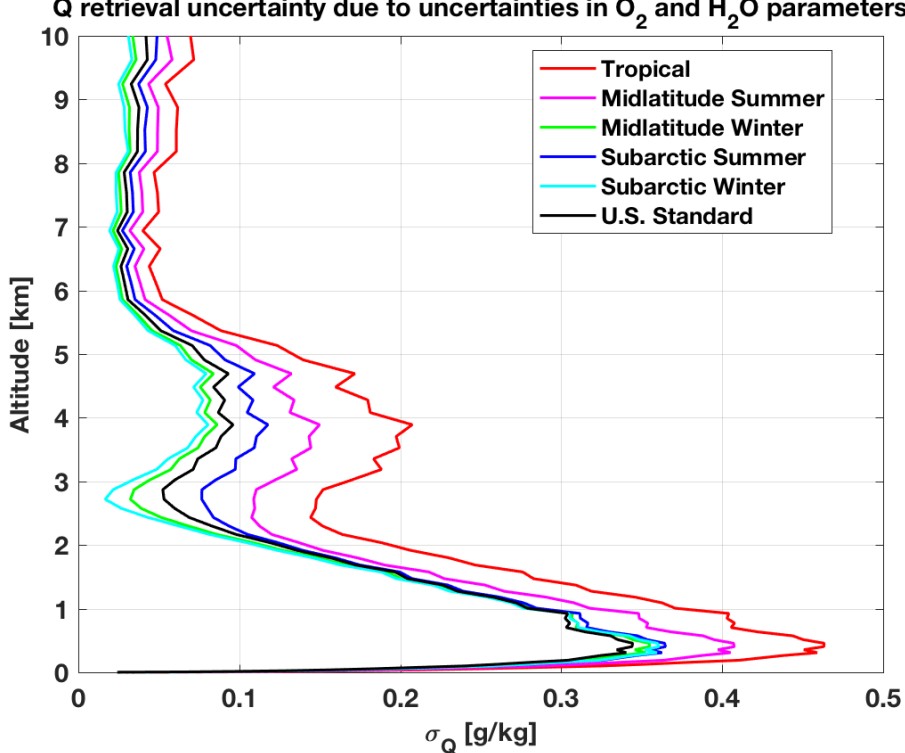

**Figure 11: As in Figure 11, but for specific humidity retrievals.**




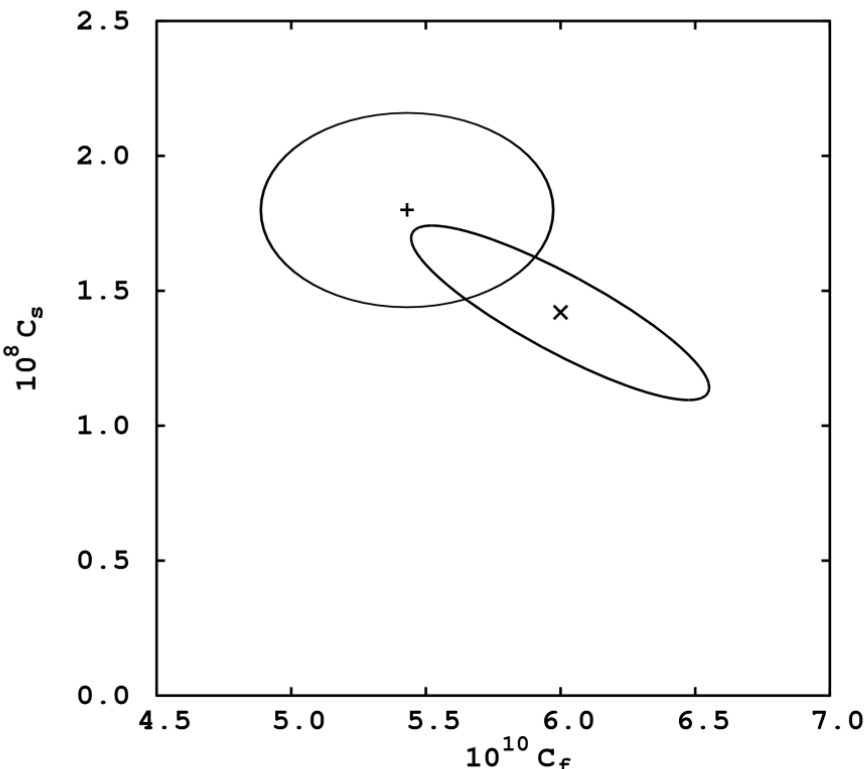

**Figure 12: Cs vs. Cf for R98 model (+) and its modification by Turner et al. (2009) (x), with uncertainty contours. Note the different scales on the two axes.**
