# Peer review of "Uncertainty of atmospheric microwave absorption model: impact on ground-based radiometer simulations and retrievals"

_Atmospheric Chemistry and Physics, 2018_

## Referee Comment (RC1) · Anonymous Referee #1 · 7 Aug 2018

Uncertainty of atmospheric absorption model in the 20-60 GHz range: impact on ground-based microwave radiometer simulations and retrievals.

The paper presents a review of current formulations of the microwave absorption parameters between 20 and 60 GHz examining in detail the uncertainties associated with each parameter. It then identifies the spectroscopic parameters that most affect microwave brightness temperature measurements at commonly used channels and it estimates the resulting covariances due to the atmospheric model.

The manuscript is well written and organized and it provides an important contribution for users of microwave data to retrieve atmospheric parameters.

My only concern is related to how these uncertainties affect the measurements and retrievals. I would expect most of the forward model uncertainties to be systematic (biases) and therefore difficult to characterize in terms of covariances? Can the authors comment on this aspect of the uncertainties.

A second question is about Fig. 11. The uncertainty in Q seems very high for Arctic atmosphere as Arctic specific humidity can be as low as 0.5 g/kg which would make the spectroscopy uncertainty about 50%. If Fig. 11 represents the uncertainty only due to the spectroscopy, to which all the other uncertainties, such as radiometric noise, etc. are added, does this make the whole microwave retrievals useless in dry conditions?

Thank you

---

## Referee Comment (RC2) · Anonymous Referee #2 · 8 Aug 2018

The paper is scientifically sound and well written. I recommend publication. I only have a very minor suggestion: Please consider adding the citation: Wentz, F., and T. Meissner, Atmospheric Absorption Model for Dry Air and Water Vapor at Microwave Frequencies below 100 GHz Derived from Spaceborne Radiometer Observations, Radio Sci., 51, doi:10.1002/2015RS005858, 2016. in the contxt of the water vapor continuum. The satellite results on the vapor continuum seems to match the gorund observation by Turner et al. 2010 pretty well.

---

## Short Comment (SC1) · 30 Aug 2018

This is an important piece of work that quantifies the total effect of spectroscopic uncertainties on ground based radiances, and vitally includes their covariance. As most spectroscopic parameters are not derived in isolation from each other this is very important to include, and to my knowledge is not, and has not been published by any other authors prior. The review of the absorption model equations with up-to-date material is particularly useful to have documented. As it is a general approach it would be really good to see how this looks for top of atmosphere geometry, though I am not suggesting the authors do this for this publication. I think it would look quite different,

probably have a lot more impact, and the 55-60 GHz impacts would not be negligible in this case. Also, it would be good to see how this looks for the whole 0-200 GHz range where I suspect in some bands the order of parameters that dominate would change. This might be limited computationally, again I don't suggest the authors do this.

One minor point is the use of the MWR to describe ground-based MW radiometers is a bit confusing, as some have already adopted the acronym for their own space-based instruments.

---

## Referee Comment (RC3) · V. Payne (Referee) · 3 Sep 2018

This paper describes an approach for quantifying impacts of spectroscopic parameters on radiative transfer model simulations and on atmospheric retrievals that accounts for correlation between different parameters. This is an important consideration which is often ignored in other studies. The paper also an example of the application of the approach to downwelling microwave radiative transfer calculations and retrievals. The work is thorough and makes a useful contribution to the body of work on quantifying retrieval uncertainties associated with spectroscopy. The paper is generally well organized and well written. I recommend publication after addressing the comments below.

[Figure]

As the authors state in the conclusions, the approach is applied to one particular widely used microwave absorption model. This should also be stated clearly in the abstract and made clear at the beginning of Section 2. Section 2 would initially seem to imply that the "review of absorption model equations" is also general, but the descriptions of the parameterizations of resonant and non-resonant absorption are particular to the MPM-based family of models. Not all atmospheric absorption models use these same parameterizations. As the authors are aware (since on page 2, within the introduction, the authors cite Long and Hodges [2012], which describes impacts of different choices of line shape parameterization on calculations of absorption for the 0.76 micron $O_2$ A-band utilized by the Orbiting Carbon Observatory and other remote sensing instruments), there are models out there for other wavelength regions that use non-Voigt line shapes for resonant absorption. Also, the description of non-resonant absorption does not apply, for example, to the widely used MT_CKD continuum model. It would seem to make sense to move the material in sub-section 2.4 up to the start of Section 2 in order to make it clear that this review of absorption model equations is not general.

Page 2, lines 50-54. The need to account for correlation between uncertainty estimates for different spectroscopic errors is general to all wavelength regions, and this is good to emphasize. The authors list a few examples of studies that discuss the impact of spectroscopic uncertainties on remotely-sensed profiles. There is one microwave example, one sub-mm example and one visible (0.76 micron) example. The authors might consider adding examples in other wavelengths. Possible examples for the thermal infrared region include Alvarado et al., [2013] and Alvarado et al [2015]. Possible examples for the near infrared region include Connor et al., [2016]. For disclosure: I happen to be a co-author on each of these particular suggested references... I am sure there are also others if you wanted to look for alternatives.

- Alvarado, M. et al., Performance of the line-by-line radiative transfer model (LBLRTM) for temperature and species retrievals: Recent updates evaluated with IASI case studies, Atmos. Chem. Phys., 13, 6687-1711 (2013)

- Alvarado, M. et al., Performance of the line-by-line radiative transfer model (LBLRTM) for temperature and species retrievals: Recent updates evaluated with IASI case studies, Atmos. Chem. Phys., 13, 6687-1711 (2013)

- Connor, B. et al., Quantification of uncertainties in OCO-2 measurements of XCO2: Simulations and linear error analysis, Atmos. Meas. Tech. Discuss., doi:10.5194/amt-2016-128, 16th June 2016

Page 3, line 81: Please make it clear that "the absorption models" means a particular set of microwave absorption models.

Page 4, line 125: Hill (1980) is a pretty old reference. Does "analogous" mean alternative Voigt parameterizations? Do the authors know if anyone has re-visited fits of line shape parameterizations to microwave experiments since then?

Page 7, lines 220-223. Does water to air mixing ratio (r'w2a) here mean that in theory the mixing between oxygen lines would be altered by the presence of water vapor? Please consider some more words here for additional clarification. Are there any calculations out there to suggest that the line mixing for oxygen should look different in wet vs dry air?

Page 12, line 1: Have these line intensities and lower state energies changed between the HITRAN 2004 and HITRAN 2016 compilations?

Page 12, line 377: Later in the paper, there is a reference for the JPL catalogue. Please also add the reference here.

Page 14, line 438-439: please add citation/reference for the 22 GHz line intensity for clarity here.

Page 16, line 500: Please provide more information here on the indirect method used in R18.

Page 25: How does the uncertainty associated with spectroscopic parameters com-

pare to the uncertainty associated with instrument noise for these examples? Please comment.

Page 28, lines 873-874: "the laboratory and field measurements". Although these are presumably referenced in Tretyakov et al. 2016, this is a very interesting point for many potential readers of this paper and therefore it would also be helpful to include the references for these laboratory and field measurements here directly also.

Typos/word choice

Page 9, line 259: "from the microwave to *the* far infrared range"

Page 10, line 303: Please consider replacing "retrieved" with "determined", since "retrieved" has its own other meaning in this context.

Page 10, line 312: Please consider replacing "involved" with "associated".

Page 10, line 313: Please consider replacing "retrieved" with "taken".

Page 11, line 342: "of which 37 **are** within the 60 GHz band, one **is** at 118 GHz and the remaining 11 **are** in the sub-mm range".
* * *

---

## Author Comment (AC1) · 12 Sep 2018

**Referee Comment 1 (RC1) by Anonymous Referee #1**

The paper presents a review of current formulations of the microwave absorption parameters between 20 and 60 GHz examining in detail the uncertainties associated with each parameter. It then identifies the spectroscopic parameters that most affect microwave brightness temperature measurements at commonly used channels and it estimates the resulting covariances due to the atmospheric model.
The manuscript is well written and organized and it provides an important contribution for users of microwave data to retrieve atmospheric parameters.

We thank the reviewer for the positive feedback.

My only concern is related to how these uncertainties affect the measurements and retrievals. I would expect most of the forward model uncertainties to be systematic (biases) and therefore difficult to characterize in terms of covariances? Can the authors comment on this aspect of the uncertainties.

We agree with the reviewer that forward model uncertainties are generally systematic, though they depend on atmospheric conditions (e.g. temperature, pressure, humidity). The presented approach aims at evaluating the total uncertainty of simulated $T_B$. In fact, spectroscopic parameters are affected by both systematic and random uncertainties. Contributions to the systematic component include theoretical assumptions and systematic experimental errors. Contributions to the random component include spread of laboratory data and experimental noise. Our analysis takes into account the total (i.e. systematic and random) uncertainty of spectroscopic parameters, which combine to result in the total uncertainty of simulated $T_B$. If parameter values are determined with methods that introduce correlation between them, their total uncertainty will also be correlated. The consideration of the parameter covariances allows to estimate more rigorously the magnitude of the total uncertainty affecting $T_B$.
We now emphasize this concept at the beginning of Section 3.

A second question is about Fig. 11. The uncertainty in Q seems very high for Arctic atmosphere as Arctic specific humidity can be as low as 0.5 g/kg which would make the spectroscopy uncertainty about 50%. If Fig. 11 represents the uncertainty only due to the spectroscopy, to which all the other uncertainties, such as radiometric noise, etc. are added, does this make the whole microwave retrievals useless in dry conditions?

We agree with the reviewer: Figure 11 shows high Q uncertainty for Arctic atmosphere. However, as stated in the manuscript (last paragraph of Section 5.2), these uncertainty profiles are to be considered just as indicative, as they depend upon the vertical grid spacing and the choice of $\mathbf{Cov}(\epsilon)$ and $\mathbf{Cov}(x_a)$. The last two terms represent the radiometric noise and the a priori uncertainty; thus $\mathbf{Cov_p}$ (whose diagonal terms are displayed in Figure 11) represents the model parameter uncertainty covariance matrix, given the choice of $\mathbf{Cov}(\epsilon)$ and $\mathbf{Cov}(x_a)$.
To present general results, we used $\mathbf{Cov}(\epsilon)$ and $\mathbf{Cov}(x_a)$ resembling typical matrices adopted in ground-based microwave profiling. The assumed a priori uncertainty for Q ($\sigma_Q$=3.2 g/kg at surface) is definitely high for Arctic conditions, and thus it dominates the gain matrix.
We have added the following sentence to Section 5.2 to make this point clear:

"Values are particularly high for relative drier climatology (e.g. Arctic); this is simply a consequence of the assumed *a priori* $\sigma_Q$, which is typical of mid-latitude climatology. Reducing $\sigma_Q$ by a factor of 10 (to be closer to values for dry climatology), the uncertainty profile would reduce roughly by the same factor."

---

## Author Comment (AC2) · 12 Sep 2018

**Referee Comment 2 (RC2) by Anonymous Referee #2**

The paper is scientifically sound and well written. I recommend publication. I only have a very minor suggestion: Please consider adding the citation: Wentz, F., and T. Meissner, Atmospheric Absorption Model for Dry Air and Water Vapor at Microwave Frequencies below 100 GHz Derived from Spaceborne Radiometer Observations, Radio Sci., 51, doi:10.1002/2015RS005858, 2016. in the contxt of the water vapor continuum. The satellite results on the vapor continuum seems to match the gorund observation by Turner et al. 2010 pretty well.

We thank the reviewer for the positive feedback.
Two more references have been added to the revised manuscript, together with the following sentence (Section A.1):
"The results of Turner et al. (2009) seem also supported by independent investigations based on satellite observations in the 10.7 to 89 GHz range (Wentz and Meissner, 2016) and around the 183 GHz line (Bobryshev et al., 2018)."

---

## Author Comment (AC3) · 12 Sep 2018

**Short Comment 1 (SC1) by Emma Turner**

This is an important piece of work that quantifies the total effect of spectroscopic uncertainties on ground based radiances, and vitally includes their covariance. As most spectroscopic parameters are not derived in isolation from each other this is very important to include, and to my knowledge is not, and has not been published by any other authors prior. The review of the absorption model equations with up-to-date material is particularly useful to have documented.

We are very grateful for this comment, which confirms our expectations that the interest on the presented analysis goes beyond the ground-based microwave radiometry community.

As it is a general approach it would be really good to see how this looks for top of atmosphere geometry, though I am not suggesting the authors do this for this publication. I think it would look quite different, probably have a lot more impact, and the 55-60 GHz impacts would not be negligible in this case. Also, it would be good to see how this looks for the whole 0-200 GHz range where I suspect in some bands the order of parameters that dominate would change. This might be limited computationally, again I don't suggest the authors do this.

The extension to upwelling $T_B$ (top-of-atmosphere geometry) and higher frequencies is currently ongoing and it will be reported in the near future.
To make sure it is clear that the same analysis would give different results at higher frequency and for upwelling $T_B$, we add the following sentence to Summary and conclusions:
"Note that the presented uncertainty covariances of spectroscopic parameters are generally valid, while the $T_B$ sensitivity analysis and uncertainty quantification are strictly valid only for the ground-based geometry and the considered frequency range"

One minor point is the use of the MWR to describe ground-based MW radiometers is a bit confusing, as some have already adopted the acronym for their own space-based instruments.

Indeed, the acronym MWR is used for space-based instruments (e.g. on board Envisat and Aquarius/SAC-D) but also for ground-based instruments (e.g. ARM units[1], MWRnet[2]). We believe that MWR should be used as the acronym for "Microwave Radiometer", with no reference to any particular type or observation geometry.
To avoid confusion with space-based instruments, we added "ground-based" and/or "downwelling" to sentences where it could have been not evident.

[1]https://www.arm.gov/capabilities/instruments/mwr
[2]http://cetemps.aquila.infn.it/mwrnet/

---

## Author Comment (AC4) · 12 Sep 2018

**Referee Comment 3 (RC3) by Vivienne Payne**

This paper describes an approach for quantifying impacts of spectroscopic parameters on radiative transfer model simulations and on atmospheric retrievals that accounts for correlation between different parameters. This is an important consideration which is often ignored in other studies. The paper also an example of the application of the approach to downwelling microwave radiative transfer calculations and retrievals. The work is thorough and makes a useful contribution to the body of work on quantifying retrieval uncertainties associated with spectroscopy. The paper is generally well organized and well written. I recommend publication after addressing the comments below.

We thank the reviewer for the positive feedback.

As the authors state in the conclusions, the approach is applied to one particular widely used microwave absorption model. This should also be stated clearly in the abstract and made clear at the beginning of Section 2. Section 2 would initially seem to imply that the "review of absorption model equations" is also general, but the descriptions of the parameterizations of resonant and non-resonant absorption are particular to the MPM-based family of models. Not all atmospheric absorption models use these same parameterizations. As the authors are aware (since on page 2, within the introduction, the authors cite Long and Hodges [2012], which describes impacts of different choices of line shape parameterization on calculations of absorption for the 0.76 micron O2 A-band utilized by the Orbiting Carbon Observatory and other remote sensing instruments), there are models out there for other wavelength regions that use non-Voigt line shapes for resonant absorption. Also, the description of non-resonant absorption does not apply, for example, to the widely used MT_CKD continuum model. It would seem to make sense to move the material in sub-section 2.4 up to the start of Section 2 in order to make it clear that this review of absorption model equations is not general.

Agreed. We added text to the abstract and the beginning of Section 2 to underline that the approach is applied to one particular microwave absorption model. We prefer not to move the whole Section 2.4 at the beginning of Section 2, as Section 2.4 mentions parameters that are defined in Section 2.1-2.3.

Abstract: "The approach is applied to a widely-used microwave absorption model (Rosenkranz, 2017) and radiative transfer calculations in the 20-60 GHz range"

Section 2: "The following Sections describe the resonant and non-resonant absorption components and the parametrization as defined in the family of absorption models considered here, i.e. R98 and R17 as well as others introduced in Section 2.4. Therefore, the review presented here applies specifically to this family of models. However, the presented approach can be considered as generally valid for any absorption model."

Page 2, lines 50-54. The need to account for correlation between uncertainty estimates for different spectroscopic errors is general to all wavelength regions, and this is good to emphasize. The authors list a few examples of studies that discuss the impact of spectroscopic uncertainties on remotely-sensed profiles. There is one microwave example, one sub-mm example and one

visible (0.76 micron) example.  The authors might consider adding examples in other wavelengths. Possible examples for the thermal infrared region include Alvarado et al., [2013] and Alvarado et al [2015].  Possible examples for the near infrared region include Connor et al., [2016].  For disclosure:  I happen to be a co-author on each of these particular suggested references... I am sure there are also others if you wanted to look for alternatives.

- Alvarado, M. et al., Performance of the line-by-line radiative transfer model (LBLRTM) for temperature and species retrievals: Recent updates evaluated with IASI case studies, Atmos. Chem. Phys., 13, 6687-1711 (2013)
- Connor, B. et al., Quantification of uncertainties in OCO-2 measurements of XCO2: Simulations and linear error analysis, Atmos. Meas. Tech. Discuss., doi:10.5194/amt-2016-128, 16th June 2016

Agreed. We now emphasize that the need for spectroscopic uncertainty characterization is general to all wavelength regions. The references above as well as Alvarado et al., 2015 have been added. Thanks for pointing this out.

Page 3, line 81: Please make it clear that "the absorption models" means a particular set of microwave absorption models.

Agreed. The sentence has been modified as "the considered microwave absorption model".

Page 4, line 125: Hill (1980) is a pretty old reference. Does "analogous" mean alternative Voigt parameterizations?  Do the authors know if anyone has re-visited fits of line shape parameterizations to microwave experiments since then?

To remove the ambiguity, the sentences have been modified as follows: "...for water vapor ($Y_i \cong 0$) (Ma et al., 2014). Then for water vapor, the line-shape function reduces to the van Vleck–Weisskopf profile: Eq(4).
The van Vleck–Weisskopf profile was demonstrated to fit experimental data well on the 22-GHz line (Hill, 1980) and the 183-GHz line (see Fig. 5 and related references from Tretyakov, 2016); also, Koshelev et al. (2018) found that speed-dependence effects amount to less than 1% deviation with respect to the van Vleck–Weisskopf profile near 22 GHz."

Page 7, lines 220-223. Does water to air mixing ratio (r'w2a) here mean that in theory the mixing between oxygen lines would be altered by the presence of water vapor?
Please consider some more words here for additional clarification.  Are there any calculations out there to suggest that the line mixing for oxygen should look different in wet vs dry air?

Correct. The mixing between $O_2$ lines should be different in dry and wet air from very general considerations. A polar water molecule as a colliding partner for an $O_2$ molecule acts significantly different from $N_2$ or $O_2$. However, no such calculations are available to our knowledge. The following sentence is added for additional clarification: "Line mixing depends on the off-diagonal elements of the collisional interaction matrix, while the diagonal elements of that matrix give the line width parameters. Therefore both mixing and broadening depend on the type of perturbing molecule; but because of the absence of either calculations or relevant measurements for r'_w2a,

the model assumes r'_w2a=r_w2a. We believe that the possible systematic impact of this assumption is smaller than other model uncertainties discussed in this paper".

Page 12, line 1: Have these line intensities and lower state energies changed between the HITRAN 2004 and HITRAN 2016 compilations?

Yes. However, R17 uses the HITRAN 2004 values. In fact, updating intensities by more accurate values will result in significant deterioration of the model if the mixing parameters are not redefined. The latter is the subject of another analysis which shall be published soon. This does not affect the work presented here, as differences between HITRAN 2016 and 2004 values are within the stated uncertainty. We added the following sentence to the revised manuscript: "Although newer calculations are available (HITRAN 2016, Gordon et al., 2017), the differences are within the assumed uncertainty."

Page 12, line 377: Later in the paper, there is a reference for the JPL catalogue. Please also add the reference here.

Done. Thanks.

Page 14, line 438-439: please add citation/reference for the 22 GHz line intensity for clarity here.

Agreed. We add the reference to HITRAN 2012 (Rothman et al., 2013) for the 22 GHz line intensity and Payne et al. (2008) for line width.

Page 16, line 500: Please provide more information here on the indirect method used in R18.

Since we believe the method used here is more rigorous then the one previously used in R18, we prefer to leave the details to that reference. However, to provide some general information, we changed the following sentence "estimated using a more indirect method in R18" to "estimated in R18, by means of an analogy with data from Payne et al. (2011)."

Page 25: How does the uncertainty associated with spectroscopic parameters compare to the uncertainty associated with instrument noise for these examples? Please comment.

Interesting question. We checked the instrument noise contribution, i.e. the diagonal terms of $\mathbf{Cov}_m$ in Eq. (39). It results of comparable magnitude (with respect to the absorption model parameter contribution), though with different vertical shape and little dependence on climatology. We have added this information at the end of Section 5.2.

Page 28, lines 873-874: "the laboratory and field measurements". Although these are presumably referenced in Tretyakov et al. 2016, this is a very interesting point for many potential readers of

this paper and therefore it would also be helpful to include the references for these laboratory and field measurements here directly also.

Correct, the references for the laboratory and field measurements are given in Tretyakov et al. 2016. Although we could report the same references here, we prefer to refer to Tretyakov et al. 2016 only, as it collects results from at least 6 different sources to produce the figure (their Figure 19) leading to the conclusion we reported.

**Typos/word choice**

Page 9, line 259: "from the microwave to *the* far infrared range"

Page 10, line 303:  Please consider replacing "retrieved" with "determined", since "retrieved" has its own other meaning in this context.

Page 10, line 312: Please consider replacing "involved" with "associated".

Page 10, line 313: Please consider replacing "retrieved" with "taken".

Page 11, line 342: "of which 37 **are** within the 60 GHz band, one **is** at 118 GHz and the remaining 11 **are** in the sub-mm range"

We thank the reviewer for spotting the above typos and making suggestions for word choice. We have accepted them all.